# Digital Economy, Technological Innovation and High-Quality Economic Development: Based on Spatial Effect and Mediation Effect

**Chenhui Ding [1], Chao Liu [2,3,\*], Chuiyong Zheng [1] and Feng Li [1]**

1 Business School, Hohai University, Nanjing 211100, China; 200213120002@hhu.edu.cn (C.D.); chyzheng@hhu.edu.cn (C.Z.); lifeng0429@163.com (F.L.)
2 School of Public Administration, Hohai University, Nanjing 210000, China
3 Jiangsu Research Base of Yangtze Institute for Conservation and High-Quality Development, Nanjing 210000, China
\* Correspondence: liu-chao@hhu.edu.cn; Tel.: +86-182-6262-8908

**Abstract:** Technological innovation and high-quality economic development are inevitable requirements of sustainable development, and the digital economy has gradually become a new engine to enhance technological innovation and the high-quality development of China's economy. Deeply discussing the effect of digital economy on high-quality economic development and clarifying the mechanism behind it can effectively grant the boosting power of digital economy to China's high-quality development, which is of great practical significance to China's sustainable economic development. In this study, the mechanism, effect, and regional heterogeneity of the impact of the digital economy on the level of high-quality economic development in 30 Chinese provinces from 2011–2019 were measured and empirically tested using a mediating effects model and a spatial Durbin model, among others. The results showed that the overall level of digital economy and high-quality development is not high, and there were both high agglomeration and low agglomeration, with obvious spatial path dependence and spatial lock-in. Digital economy could promote the high-quality development level of the economy, and the spatial spillover effect was remarkable. In addition, the function of digital economy in promoting high-quality economic development in the eastern, central, and western regions was gradually weakened. Besides, the technological innovation was an important transmission path of digital economy to high-quality economic development. Based on these findings, it is proposed that decision-makers should strengthen digitalization efforts so that the digital economy can become a powerful tool to narrow the digital divide. Further, the dynamic and differentiated digital economy development strategy should be implemented to reduce regional development imbalances in an effective manner.

**Keywords:** digital technology; technological innovation; high-quality economy; spatial effects; mediating effects

## 1. Introduction

Digital economy has advantages in technological innovation ability, industrial integration ability, and market expansion ability [1], and also plays an important role in promoting high-quality economic development. It is considered to be an important driving force to promote this development. In particular, in recent years, the Chinese government has continuously increased its investment in the digital economy and successively issued a number of major plans and policy guidelines regarding the digital economy. At the central economic work conference in 2019, it proposed to take "focus on promoting high-quality development and vigorously developing the digital economy" as it's basic task, which has determined the direction for the transformation of China's economic growth mode. In the 14th five-year plan, it was clearly emphasized that "accelerating the development of digital economy is the practical need to build a new engine of high-quality development". The

white paper on the development of China's digital economy (2021) issued by the Chinese Academy of Information and Communications pointed out that in 2020, the scale of China's digital economy reached 39.2 trillion yuan, accounting for 38.6% of the GDP, and its growth rate exceeded the growth rate of the GDP by more than three times, indicating that the scale of digital economy has gradually become the core growth pole and development engine of high-quality development of the national economy while maintaining a high-speed growth trend [2]. However, the function of digital economy on high-quality economic development is not clear. With the background of the economy entering the new normal, it has become an urgent problem to seek new growth momentum, leading China's economy from the "shift" of high growth rate and low-quality mode to the high-quality stage driven by innovation, and seeking a solution to smoothly turn onto the track of high-quality economic development.

Although the digital economy is becoming an important part of the national economy, theoretical research on this topic lags compared to the rapid development at the practical level and the considerable attention at the policy level. Digital economy is a powerful kinetic energy used by the state to release the efficiency of digital innovation, improve the rate of technological innovation, and promote the upgrading of industrial structure, thus accelerating the economic construction from "quantity growth" to "quality growth" (Clifton et al., 2019). However, there is a paucity of systematic and theoretical researches on the digital economy. Previous studies have argued that the digital economy can influence high-quality economic development from a multi-dimensional perspective which includes the macro, meso, and micro levels [3,4]. The micro-level studies are mainly analyzed using economic concepts such as economies of scale, economies of scope, long-tail effect, and matching efficiency [5]. The meso-level studies are primarily focus on the technological upgrading of the digital economy and the technological diffusion to other industries [3,6]. The macro-level studies are mainly researched on the role of the digital economy in high-quality development from the perspective of economic factors, allocative efficiency, and productivity [7,8]. Besides, scholars have also launched studies related to the fields involved in the above references such as the influence of the digital economy on inclusive growth [8] and the relationship between the Internet and economic growth [9–12] on Total Factor Productivity [13–17]. However, the path through which the digital economy is to contribute to the high-quality development of the economy has not been clearly answered by previous studies, which provides a marginal contribution opportunity for this study.

In summary, is the digital economy driving China's high-quality economic development? How does it drive this development, and what is the mechanism among its operation? What are the patterns and characteristics of the digital economy's impact on economic quality, and how does it vary spatially? These questions have become important issues that cannot be avoided in the digital economy era. The study of the above questions and a comprehensive evaluation of the role of the digital economy on the development of a high-quality economy can provide a basis and reference for the decision making of relevant institutions, which is of practical significance for the formulation and implementation of relevant policies, and provides theoretical and empirical references for the promotion of the digital economy in China. Therefore, this study drew on the existing literature and constructed an indicator system to measure the digital economy and f the high-quality economy using data from 30 Chinese provinces from 2011–2019; as well, a theoretical analysis framework of the digital economy and high-quality economic development from the perspective of technological innovation was developed. Various econometric methods such as mediating effects and the spatial Durbin model were adopted to test the impact of the digital economy on high-quality economic development. The results confirmed that the digital economy significantly contributes to the development of a high-quality economy with spatial spillover characteristics, and that technological innovation was the main path and mechanism among this contribution.

The contributions of this study are as follows. First, there is little research that integrates digital technology, technological innovation, and high-quality economic develop-

ment into the same research framework. This article innovatively combined the three to deepen the existing literature and research. Second, within this framework, this article attempted to answer the question of how digital technology mainly affects economic development and clarified the influence of technological innovation on the path of digital technology's impact on high-quality development, which is of great theoretical and practical significance. Third, it attempted to supplement the shortage of empirical methods for research in this field. The impact of the digital economy on the provincial level was focused, and the temporal and spatial evolution characteristics and influence relationship between the digital economy and high-quality development of the economy were discussed in a more detailed way.

The remainder of this paper is formed as follows. The second section provides a systematic review of the existing literature. The third section presents the hypotheses and tests, and analyzes the direct mechanism, indirect mechanism, spatial spillover effect, and technological innovation mediating effect of the digital economy on high-quality economic development. The fourth section presents the research design, including the construction of the model and description of variables. The fifth section presents the empirical tests, including benchmark, regional heterogeneity, and spatial effects tests. The sixth section presents the robustness test. The last section presents the conclusion and research implications.

## 2. Literature Review

At present, the digital economy, technological innovation, and high-quality economic development are attracting increased scholarly attention. However, the relationship between the three has not yet been determined, and the study of whether the digital economy and technological innovation can become an important driving force for China's high-quality economic development is still controversial. Therefore, it is of great practical significance to examine this impact in greater detail.

Scholarly work on the relationship between these aspects can be roughly divided into the following three categories. The first category is researches related to the digital economy and high-quality development. Scholars believe that the digital economy can promote high-quality economic development by enhancing the quality of certain factors [18]; as well, through the evolutionary differences in factor allocation changes, the evolution of industrial upgrading drivers, and the evolution of economic growth quality [19]. Concurrently, the high growth, strong diffusion, and cost reduction characteristics of the digital economy due to the incremental marginal gains of information will also promote high-quality economic development through external performance and internal dynamics [20]. Some scholars have also studied the relationship between the two through empirical evidence. For example, Zhao (2020) verified that the digital economy can significantly promote high-quality economic development by measuring two variables across 222 cities in China at the prefecture level and above [21], and pointed out that entrepreneurial activity is an important mechanism for the digital economy to disseminate the dividends of high-quality economic development [21]. Zhang (2019) confirmed that digital finance could promote the inclusive growth of China's economy in the context of the digital economy using the country's Digital Inclusive Finance Index and Household Tracking Survey data [8]. Through an empirical analysis, Zhang et al. (2021) found that the digital economy can significantly drive the quality of economic growth in the eastern and western regions of China [12]. The second category is researches on technological innovation and high-quality economic development. By constructing an input–output model that includes the digital economy, Wang & Wu (2020) confirmed that the development of China's digital economy could significantly improve social production efficiency and promote economic development [22]. The specific mechanisms include the continuous improvement and promotion of new infrastructure construction related to, and the continuous extension and expansion of the breadth and depth of, integration of the digital economy with the traditional real economy, as well as the continuous development of new industries and new business models generated

by the digital economy [22]. The key drivers of high-quality economic development are scientific discovery, technological invention, and industrial innovation [23]. The promotion of core technological innovation is conducive to promoting high-quality economic development in China [24]. Scholars believe that technological innovation can promote high-quality economic development [25] and regard technological innovation as one of the main drivers of economic growth [26,27]. However, their conclusions are inconsistent, and some scholars believe that the impact of technological innovation on economic growth is not significant [28]. The third category is researches related to the digital economy and technological innovation. Based on the Cobb–Douglas production function and vector autoregressive model, Xiong and Cai (2020) conducted an empirical analysis of the impact of digital economy-driven innovation in Yangtze River Delta city clusters. They found that the digital economy can positively contribute to the improvement of technological innovation and product innovation [29]. Jun et al. (2020) empirically analyzed the impact mechanism of the digital economy on regional innovation at the enterprise, industry, and regional levels by constructing an evaluation system of the digital economy's level of development at the provincial level in China and found that the development of the digital economy has a significant effect on promoting the improvement of regional innovation capacity [30].

In summary, given that the relationship between the digital economy and high-quality development has been proposed relatively recently, there is no unified measurement standard, and the relevant literature is limited. Most existing studies combine the three, and no detailed analysis has been conducted on the impact of mechanisms and effects on the digital economy, technological innovation, and economic quality. Based on the existing research, this study reexamined the mechanism and effect of digital economy on high-quality development, and discussed the role of technological innovation between them, so as to further deepened the understanding of the impact mechanism of digital economy on high-quality economic development. An in-depth study of the impact of digital economy on economic quality can not only fully tap the value of digital economy in improving economic quality and efficiency, but can also provide theoretical support for China to promote the development of digital economy. At the same time, it is of great practical significance to promote the implementation of innovation-driven strategy, strengthen technological innovation, and jointly drive high-quality economic development with digital economy and technological innovation.

## 3. Theoretical Analysis and Research Hypotheses

In previous studies, most of the three are combined. Few authors carry out systematic and in-depth theoretical analysis and quantitative research on the integration of digital economy, technological innovation, and economic development within the same research framework. However, in the Chinese context, the relationship between the three is becoming closer and closer and, therefore, it is necessary to clarify the internal logic and theoretical basis of the three. The digital economy can exert a direct impact on high-quality economic development and an indirect impact through technological innovation. In addition, the digital economy may also have a spatial spillover effect on the development of economic quality under the combined effect of Moore's Law and Metcalfe's Law [21]. Based on the above, this study examines and demonstrates the impact of the digital economy on the high-quality development in terms of its influence mechanism and effects and proposes the following hypotheses.

### 3.1. Direct Impact Mechanism of Digital Economy on High Economic Quality

The digital economy relys on the strong diffusion based on digital technology and new infrastructure. Thus it can give full play to the multiplier effect of data elements on the efficiency of other production factors, release the new vitality of economic development, improve total factor productivity, and promote high-quality economic development [18]. In addition, digital economy can improve the quality of advanced elements to achieve the

purpose of promoting high-quality economic development [12,13]. The evolution driven by industrial upgrading and economic growth quality promote high-quality economic development [14]. At the same time, the characteristics of digital economy such as high growth, strong diffusion, and cost reduction due to the increasing marginal income of information will also promote high-quality economic development through "external performance" and "internal driving force" [15]. Therefore, this paper puts forward the following assumptions:

**Hypothesis H1 (H1).** *Digital economy plays a direct and positive role in promoting high-quality development.*

### 3.2. Indirect Impact Mechanism of Digital Economy on High-Quality Economic Development

Digital economy, the influencing mechanism between technological innovation and high economic quality, can be described from the following aspects: (1) the role of digital economy in technological innovation. As a highly knowledge- and technology-intensive economic form, the rapid development of digital economy is very important to promote technological innovation [31]. Industries corresponding to digital economy usually have high knowledge density and strong innovation ability [18]. In addition, the strong diffusion attribute of digital economy can promote traditional industries to continuously absorb new technologies of digital economy and apply them to organizational management, production, marketing, and other processes, so as to realize innovation in organizational management, production, and marketing and improve organizational operation efficiency. At the same time, when traditional industries are integrated with digital economy, the development and utilization of new processes, methods, and products improve the innovation of the economy. (2) The role of technological innovation in high-quality development. The key drivers of high-quality economic development are scientific discovery, technological invention, and industrial innovation [21]. Scientific and technological innovation is regarded as one of the main driving forces driving economic growth [19,20]. It is believed that promoting core technological innovation is conducive to promoting high-quality economic development in China [12,14], and its approaches mainly include industrial structure upgrading, energy conservation and emission reduction, achievement sharing, etc. (3) Digital economy plays a role in high-quality development through technological innovation. Digital economy infiltrates all links of social and economic development in the form of a general technology [32], which requires a large amount of innovative technology support. The obvious advantages of digital economy in technological innovation, industrial integration, and market expansion [33] show its important role in promoting high-quality economic development. Technological innovation has improved all aspects of economy and society, whether as consumer goods or intermediate goods, the improvement of its value is accompanied by the realization of technological innovation. Technological innovation has become the key aspect of digital economy in promoting high-quality development. Digital economy penetrates the economy through technology, brings technological innovation, and further promotes high-quality development through innovation.

**Hypothesis H2 (H2).** *Digital economy promotes technological innovation.*

**Hypothesis H3 (H3).** *Technological innovation promotes high-quality economic development.*

**Hypothesis H4 (H4).** *Digital economy can enable high-quality economic development by improving technological innovation capability.*

### 3.3. Spatial Spillover Mechanism of Digital Economy to High-Quality Economic Development

The digital economy has two important characteristics: compressing the spatial and temporal distance and enhancing the breadth and depth of inter-regional economic activity linkages. Related studies conducted in the Chinese context [34,35] similarly support the

conclusion that the Internet has spatial spillover effects. Economic activities in cities are also significantly spatially related, with the Internet generating spatial spillover effects on regional economic development in terms of economic growth [36–38], resource mismatch [39], and digital finance [40]. Thus, it is logical that the impact of the digital economy on high-quality development should also have spatial spillover effects, as shown in Figure 1. Therefore, the following research hypothesis is proposed:

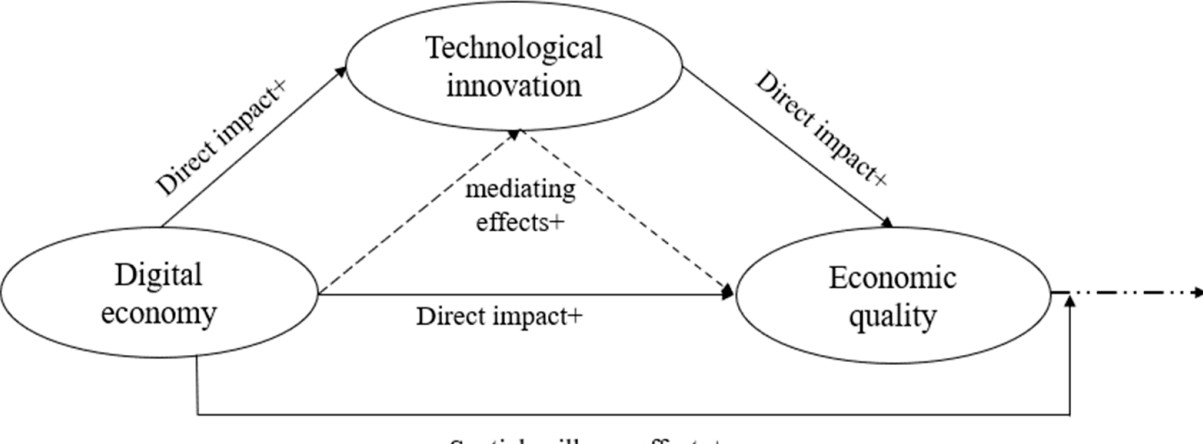

**Figure 1.** Conceptual framework.

**Hypothesis H5 (H5).** *Digital economy can affect the high-quality economic development level of adjacent areas through spatial spillover effect.*

In conclusion, the theoretical framework shown in Figure 1 can be summarized by summarizing the assumptions of this study on the relationship between digital economy, technological innovation, and high economic quality.

## 4. Methodology and Materials

In previous studies, few authors have carried out systematic and in-depth theoretical analysis and quantitative research on digital economy, technological innovation, and economic development. However, in the Chinese context, the relationship between the three is becoming closer and closer. It is necessary to clarify the internal logic and theoretical basis of the three. In order to fully explore the impact of digital economy on high-quality economic development, this paper uses the intermediary model and spatial model to discuss the direct, indirect, and spatial spillover effects of digital economy on high-quality economic development, and explores the important role of technological innovation. The specific research methods, index selection, data sources, and data processing methods are as follows.

### 4.1. Model Construction

In order to verify the above research assumptions, the following basic models are constructed for the direct transmission mechanism.

$$Ehqd_{it} = \alpha_0 + \alpha_1 Dige_{it} + \alpha_j Control_{it} + \mu_i + \varepsilon_{it} \tag{1}$$

In Equation (1), $Ehqd_{it}$ represents the level of high-quality economic development in province $i$ in period $t$. $Dige_{it}$ indicates the level of digital economy development in province $i$ in period $t$. The vector $Control_{it}$ represents a set of control variables. $\mu_i$ denotes the individual fixed effects of province $i$ that do not vary over time, and $\varepsilon_{it}$ denotes the random disturbance term.

In addition to the direct effect embodied in Equation (1), in order to discuss the possible mechanisms of action of the digital economy on the high-quality development

of the economy, another verification is conducted to determine whether technological innovation is a mediating variable between the two. The specific test steps are as follows. On the assumption that *Dige* passes the significance test of the coefficient $\alpha_1$ of the linear regression model (1), the linear regression equation for *Dige* on the mediating variable of technological innovation (*Tc*) and the linear regression equation for *Dige* and the mediating variable *Tc* on *Ehqd* are constructed. The significance of the regression coefficients of $\beta_1$, $\gamma_1$, and $\gamma_2$ are used to determine the existence of a mediating effect. The specific formulas for the above regression models are as follows.

$$Tc_{it} = \beta_0 + \beta_1 Dige_{it} + \beta_c Control_{it} + \mu_i + \varepsilon_{it} \tag{2}$$

$$Ehqd_{it} = \gamma_0 + \gamma_1 Dige_{it} + \gamma_2 Tc_{it} + \gamma_c Control_{it} + \mu_i + \varepsilon_{it} \tag{3}$$

Finally, to further discuss the spatial spillover effects of the digital economy on high-quality development, the spatial interaction terms of these two and other control variables are introduced to Equation (1), which is further extended into a spatial panel econometric model:

$$Ehqd_{it} = \alpha_0 + \rho WEhqd + \phi_1 WDige_{it} + \phi_c WControl_{it} + \alpha_1 Dige_{it} \\ + \alpha_j Control_{it} + \mu_i + \delta_t + \varepsilon_{it} \tag{4}$$

where $\rho$ represents the spatial autoregressive coefficient and $W$ represents the spatial weight matrix. $\phi_1$ and $\phi_c$ represent the elasticity coefficients of the core explanatory variables as well as the spatial interaction terms of the control variables. Equation (4) is the spatial Doberman model (SDM), which includes the spatial interaction terms of explained variables and explanatory variables. To analyze the impact of the digital economy on regional innovation performance in a more objective manner, the geographic weight matrix, spatial weight matrix of economic characteristics, and Moran's index (Moran's *I*) are constructed to measure spatial correlation.

The spatial weight matrix (*W*) is the key to performing the spatial correlation test. Due to space limitations, only the definitions of the distance matrix and economic-distance weight matrix are shown here.

(1)    Geographical matrix

$$W_{ij} = \begin{cases} 1/d_{ij}, & i \neq j \\ 0, & i \neq j \end{cases} \tag{5}$$

In Equation (5), $d_{ij}$ denotes the linear distance between province *i* and the capital city of province *j*.

(2)    Economic-distance spatial matrix

$$w_{ij} = \begin{cases} 1/\left|\overline{Y}_i - \overline{Y}_j\right|, & i \neq j \\ 0, & i = j \end{cases} \tag{6}$$

where $\overline{Y}_i = \frac{1}{T-T_0} \sum\limits_{T=t_0}^{T} Y_{it}$, $\overline{Y}_i$ denotes the average value of real GDP per capita in region *i*, as shown in Equation (6).

(3)    Spatial autocorrelation test

A spatial correlation test of the observations is required when applying the spatial econometric model. Moran's *I* is the earliest method used to test spatial correlation, which can reflect the degree of spatial association of observations. It can test whether there is a

similar, dissimilar, or independent relationship between neighboring regions in a spatial system. The global Moran's *I* is calculated as follows:

$$I = \frac{n \sum_{i=1}^{n} \sum_{j}^{n} W_{ij}(x_i - \overline{x})(x_j - \overline{x})}{\sum_{i=1}^{n} \sum_{j=1}^{n} (x_i - \overline{x})^2} = \frac{n \sum_{i=1}^{n} \sum_{j \neq 1}^{n} W_{ij}(x_i - \overline{x})(x_j - \overline{x})}{S^2 \sum_{i=1}^{n} \sum_{j=1}^{n} W_{ij}} \tag{7}$$

Here, $\sum_{i=1}^{n}(x_i - \overline{x})^2/n$, $\overline{x} = \sum_{i=1}^{n} x_i/n$. $x_i$ and $x_j$ represent the observations of regions *i* and *j*, respectively; *n* represents the total number of regions; $W_{ij}$ represents the weight matrix, built based on different criteria. The value of Moran's *I* is between −1 and 1. A statistic value greater than zero indicates that observations with similar attribute spatial units are spatially positively correlated. On the contrary, a statistical value less than zero indicates that observations with similar attribute spatial units are spatially negatively correlated.

*4.2. Measurement and Description of Variables*

(1) Explained variables. High quality economic development is in the period of economic structure transformation. China's economic growth will change from its traditional extensive economic growth to a more efficient, high-quality, economic, environmentally friendly, and green economic growth model, and the economic growth under this model is also high-quality economic development. Referring to the studies of Ren & Yang (2020), Guo & Chen (2021), and Ge & Wu (2021), nominal GDP per capita is transformed into real GDP per capita, and 2011 was chosen as the base period [25,41,42]. Real GDP per capita was selected as the explained variable to reflect the level of high-quality economic development in different regions, denoted as Ehqd.

(2) Explanatory variables. Digital economy refers to a series of economic activities with the use of digital knowledge and information as key production factors, a modern information network as an important carrier, and the effective use of information and communication technology as an important driving force for efficiency improvement and economic structure optimization. The application of the Internet, cloud computing, big data, Internet of things, financial technology, and other new digital technologies in the process of information collection, storage, analysis, and sharing has changed the method of social interaction. Based on this, and drawing on the idea of Liu et al. (2020) of using Internet development as the basis of measurement and adding a digital transaction indicator system, this study combines the availability of relevant data to measure the comprehensive development index of the digital economy in two dimensions, namely Internet development and digital financial inclusion. The Internet is the carrier and bedrock of the development of the digital economy [43–45]. To measure the level of Internet development at the provincial level, this study draws on Huang (2019) to focus on four aspects: Internet penetration rate, personnel employed in related industries, the output of related industries, and mobile phone penetration rate, which are measured using the number of Internet users per 100 people, the proportion of personnel employed in computer services and software industries to those employed in other urban industries, the total value of telecommunications business per capita, and the number of mobile phone users per 100 people, respectively [46]. Digital financial inclusion is an important manifestation of the development of the digital economy. This variable is measured using the provincial digital financial inclusion index in China, compiled byGe et al. (2021), which measures digital financial coverage, depth of use, and degree of digitization [42]. Ultimately, drawing on Zhao et al. (2020), the comprehensive digital economy development index, denoted as Dige, is obtained by standardizing and then downscaling the data of the above five indicators through principal component analysis [21].

(3) Mediating variables. Technological innovation (Tc) is an important source of new dynamic energy in the new era. Technological innovation is all the activities that innovators use, as well as new technologies and inventions which change production factors and production conditions, and commercialize the change results. Referring to the studies of

Shangguan (2016) and Xiong & Wang (2020), the market technology turnover is selected as a proxy variable [47,48], logarithmically processed, and denoted by Tc.

(4) Control variables. Government intervention (Gov) is measured by the share of local fiscal expenditures in the local GDP, which measures the intensity of government intervention in the economy in each region. Industrialization level (Ind): industrialization results in rapid economic growth with increased adverse effects on the green development process. This study uses the ratio of secondary industry output to regional GNP as a proxy for Ind. Foreign direct investment (FDI): regions continue to attract foreign investment, which may be an active contributor to the "wave phenomenon" and have a greater impact on the stability of economic performance. The share of FDI in regional GDP is used to measure each region's dependence on foreign investment and is logarithmized. The level of urbanization (Cin) is expressed by the urbanization index. Level of human capital (Human): high-skilled and innovative talent plays an important role in China's high-quality economic development, and the accumulation of human capital provides this talent to support high-quality economic development. This study uses the proportion of students enrolled in colleges and universities in each region to the total number of students in the region as a proxy for the level of human capital. The degree of marketization (Market): a higher degree of marketization reflects the ability of the market to regulate resource allocation, and a higher degree of marketization can influence the level of economic quality by reducing the distortion of resource allocation and enabling innovation factors to be reasonably allocated according to the relevant market signals. Thus, this study adopts the marketization index to measure the degree of marketization.

### 4.3. Data Sources and Descriptive Statistics

Given the availability of digital financial inclusion data and the lack of data from Hong Kong, Macao, Taiwan, and Tibet, we selected the panel data of 30 provinces in China from 2011 to 2019 for research. The data are winsorized on the quantile of 1~99% to avoid outliers The indicators mentioned in the article that use currency as the unit of measurement are deflated to ensure the accuracy and credibility of the regression results, taking 2011 as the base year. In addition, the missing data in the research are interpolated. The data used in this study are mainly from the China Statistical Yearbook, China Statistical Yearbook on Science and Technology, and Guotaian, among which the digital economy data are from the China Statistical Yearbook on Science and Technology EPS Global Database and the China Provincial Digital Financial Inclusion Index. The results of the digital economic index are shown in Figure 2. Due to space constraints, only the values of the East, middle, and West indexes and the average value of each province are displayed. Among them, the digital economy index of the eastern region is much higher than that of the central and western regions.

Descriptive statistics are utilized for all the variables tested in the econometric model of this study, which are presented in Table 1. The results indicate that the mean level of economic development is 1.26, the maximum value is 2.252, and the minimum value is 0.497. This indicates that the quality of economic development varies widely among different regions, which is consistent with the results of previous studies. There are also significant differences in the remaining variables.

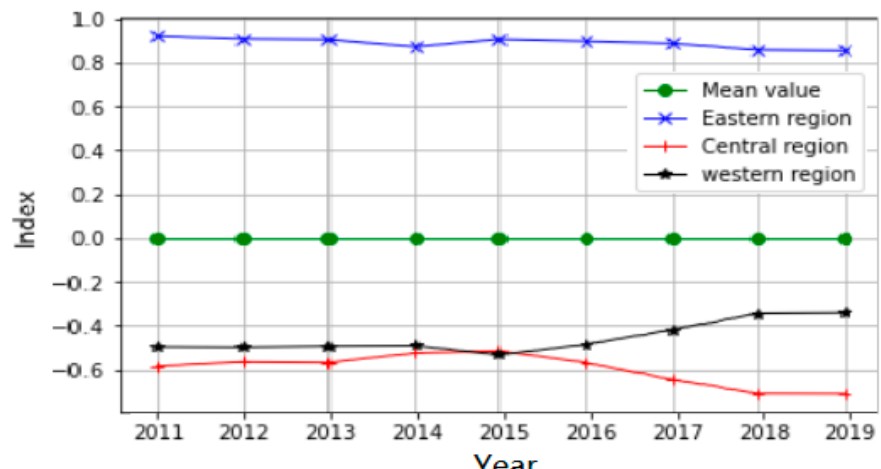

**Figure 2.** China's digital economy development index from 2011 to 2019. Note: the eastern region of China includes Beijing (BJ), Tianjin (TJ), Hebei (HB), Liaoning (LN), Shanghai (SH), Jiangsu (JS), Zhejiang (ZJ), Fujian (FJ), Shandong (SD), Guangdong (GD), and Hainan (HI); the central region includes Shanxi (SX), Jilin (JN), Heilongjiang (HL), Anhui (ah), Jiangxi (JX), Henan (HN), Hubei (HU), and Hunan (HA); the western region includes inner Mongolia (nm), Guangxi (GX), Chongqing (CQ), Sichuan (SC), Guizhou (GZ), Yunnan (YN), Shaanxi (SA), Gansu (GS), Qinghai (QH), Ningxia (NX), and Xinjiang (XJ).

**Table 1.** Descriptive statistics of variables.

|  |  | Obs | Mean | Std | Min | Max |
|---|---|---|---|---|---|---|
| Explained variables | Ehqd | 270 | 1.569 | 0.452 | 0.692 | 2.827 |
| Explanatory variables | Dige | 270 | 0 | 0.985 | −1.025 | 3.62 |
| Mediating variables | Tc | 270 | 4.623 | 1.742 | 0.357 | 8.409 |
| Control variables | Market | 270 | 1.859 | 0.314 | 0.936 | 2.392 |
|  | FDI | 270 | 0.13 | 1.477 | −7.652 | 1.997 |
|  | Human | 270 | 9.204 | 0.896 | 7.609 | 12.502 |
|  | Cin | 270 | 57.642 | 12.159 | 36.796 | 89.328 |
|  | Ind | 270 | 0.342 | 0.079 | 0.118 | 0.51 |
|  | Gov | 270 | 24.858 | 10.213 | 11.89 | 61.211 |

## 5. Empirical Test on the Impact of the Digital Economy on High-Quality Economic Development

### 5.1. Analysis of Basic Estimation Results

Table 2 shows the estimation results of the impact of the digital economy on high-quality economic development. The fixed-effects model is selected based on the Hausman test. The estimated coefficient of the digital economy index in the benchmark regression and mediating effects (Ehqd1) is significantly positive, indicating that the digital economy significantly contributes to high-quality economic development at the inter-provincial level, thus verifying hypothesis H1. For the control variables, the coefficient of foreign trade dependence has a significant negative correlation, indicating that the entry of foreign capital makes local enterprises prone to technological dependence as well as the crowding out of local technology, which is not conducive to the improvement of technological innovation capacity nor the quality of the economy [49]. The coefficient of government intervention is negative and significant, indicating that excessive government intervention may not promote the high-quality development of the economy. The coefficient of the

industrialization level is positive and significant, indicating the importance of the increase in industrialization in improving the quality of regional economic development. The coefficient of urbanization is positive but insignificant, indicating that the expansion of the size of the urban economy is detrimental to the improvement of the quality of inter-provincial economic growth, which is consistent with the results of Zeng et al. (2019) [50]. The coefficient of human capital is positive but not significant, indicating that, although China's human capital can contribute to the high-quality development of the economy to an extent, the appropriate function is not well realized. The coefficient of the marketization index is positive and significant, which indicates that a higher level of marketization helps in the allocation of market resources, thus positively impacting the high-quality development of the inter-provincial economy.

**Table 2.** Estimates of the impact of the digital economy on high-quality economic development.

| Variables | Benchmark Regression | Mediating Effects | | |
|---|---|---|---|---|
| | Ehqd | Ehqd1 | Tc | Ehqd2 |
| Dige | | 0.082 *** | 0.407 * | 0.077 *** |
| | | (3.66) | (1.80) | (3.41) |
| Tc | | | | 0.014 ** |
| | | | | (2.03) |
| FDI | −0.072 *** | −0.071 *** | −0.136 | −0.069 *** |
| | (−4.00) | (−4.10) | (−0.78) | (−4.02) |
| Gov | −0.021 *** | −0.019 *** | 0.015 | −0.019 *** |
| | (−11.76) | (−11.02) | (0.83) | (−11.21) |
| Ind | 0.009 *** | 0.009 *** | −0.016 | 0.009 *** |
| | (7.22) | (7.28) | (−1.31) | (7.49) |
| Cin | 0.003 | 0.001 | 0.082 *** | 0.000 |
| | (1.40) | (0.57) | (3.42) | (0.09) |
| Human | 0.050 | 0.048 | 0.730 * | 0.037 |
| | (1.27) | (1.24) | (1.90) | (0.97) |
| Market | 0.017 ** | 0.023 *** | 0.183 ** | 0.020 *** |
| | (2.25) | (3.01) | (2.38) | (2.66) |
| Constant | 1.155 *** | 1.213 *** | −2.063 | 1.242 *** |
| | (6.75) | (7.27) | (−1.23) | (7.48) |
| N | 270 | 270 | 270 | 270 |
| Adj-R2 | 0.62 | 0.64 | 0.51 | 0.65 |

t-statistics or z-statistics in parentheses; *** $p < 0.01$, ** $p < 0.05$, * $p < 0.1$.

The impact mechanism of technological innovation was also verified with the mediating effects model. As shown in Table 2, the regression coefficients of the technological innovation and digital economy index in the mediating effects (Ehqd1) and the mediating effects (Ehqd2) are positive and significant. Among them, the regression coefficient of the digital economy on technological innovation in the mediating effects (TC) is positive and significant, which supports hypothesis H1, in which the digital economy could significantly enhance technological innovation. After adding the mediating variable of technological innovation in the mediating effects (Ehqd2), the coefficient of the impact of the digital economy and high-quality economy is also significantly positive. However, there is a decrease in its coefficient compared to the mediating effects (Ehqd1), indicating that the digital economy significantly promotes high-quality economic development. The result supports hypotheses H3, H1, and H4. After performing stepwise regression, the Sobel test and bootstrapping test were conducted to ensure the reliability of the results. The results reveal that $p = 0.054 < 0.1$, indicating that the tests significantly rejected the original

hypothesis on the absence of mediating effects at the 10% level. In summary, technological innovation is an indirect impact mechanism of the digital economy to promote high-quality economic development, which also validates hypothesis H4.

### 5.2. Analysis of Spatial Spillover Effects

A spatial correlation between the digital economy and high-quality economy should be verified before conducting the spatial effect analysis. The main methods employed in testing spatial autocorrelation are Moran's *I* and the Cetis-Ord index, which are widely used index methods. The value of Moran's *I* of the digital economy and high-quality economy each year were calculated under the Geographic Matrix(W1), proximity weight matrix(W2) and economic-distance matrix(W3), as shown in Table 3. It is observed that the Moran's *I* of high-quality economy and digital economy development levels from 2011 to 2019 under the different matrix passed the significance test at the 5% and 1% levels. The results indicated that the digital economy and high-quality economy levels of each Chinese province had significant spatial clustering characteristics, and there was a positive correlation in spatial dependence.

**Table 3.** Moran's *I* of the development levels of high-quality economy and digital economy from 2011 to 2019.

| Year | Ehqd | | | Dige | | |
|---|---|---|---|---|---|---|
| | **W1** | **W2** | **W3** | **W1** | **W2** | **W3** |
| 2011 | 0.149 *** | 0.451 *** | 0.849 *** | 0.078 *** | 0.276 *** | 0.276 *** |
| 2012 | 0.146 *** | 0.442 *** | 0.848 *** | 0.074 *** | 0.257 *** | 0.257 *** |
| 2013 | 0.143 *** | 0.435 *** | 0.855 *** | 0.054 ** | 0.212 ** | 0.212 ** |
| 2014 | 0.137 *** | 0.424 *** | 0.860 *** | 0.055 ** | 0.194 * | 0.194 *** |
| 2015 | 0.133 *** | 0.421 *** | 0.826 *** | 0.076 *** | 0.287 *** | 0.287 *** |
| 2016 | 0.129 *** | 0.43 *** | 0.831 *** | 0.063 *** | 0.229 *** | 0.229 *** |
| 2017 | 0.134 *** | 0.452 *** | 0.695 *** | 0.04 ** | 0.204 * | 0.204 ** |
| 2018 | 0.105 *** | 0.381 *** | 0.566 *** | 0.038 *** | 0.212 * | 0.212 ** |
| 2019 | 0.132 *** | 0.422 *** | 0.861 *** | 0.04 *** | 0.213 ** | 0.250 *** |

Notes: *** $p < 0.01$, ** $p < 0.05$, * $p < 0.1$.

Further, the spatial distribution of hot and cold regions of China's digital economy was measured by the GETIS ord index (g *). When g * > 0, it indicates that the value around area I is high and is a hot spot area; on the contrary, it can be said that it belongs to the cold spot area. With the help of the tool ArcGIS 10.7, the spatial distribution map of China's digital economy cold and hot spots from 2011 to 2019 was calculated, as shown in Figure 3. In terms of the distribution map that the digital economy, hotspots are concentrated in the more developed areas along the eastern coast, while cold spots are mostly distributed in the central, western, and northeast regions. Based on the development trends in 2011, 2014, 2016, and 2019, it is observed that hot spots tended to spread to the east, and cold spots tended to spread to the northwest.

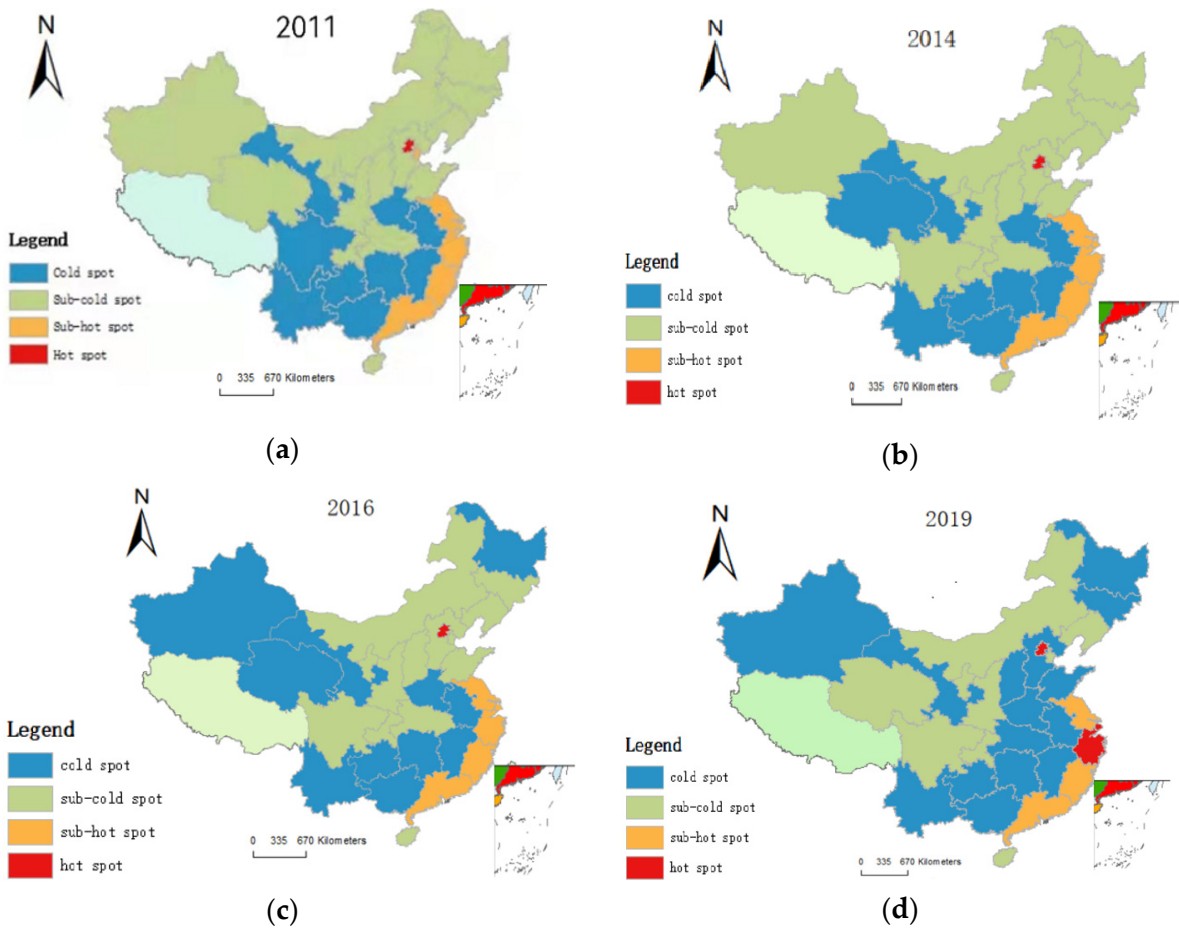

**Figure 3.** Spatial distribution map of cold and hot spots of China's digital economy from 2011 to 2019, which (**a**) 2011; (**b**) 2014; (**c**) 2016; (**d**) 2019.

The spatial distribution map of cold and hot spots for the high-quality economic development level in China is shown in Figure 4. It can be seen that the hot and cold spots' distribution of high-quality economic development and digital economy were very similar to each other in 2011, 2014, 2016, and 2019. Further, the hot spots were concentrated in the eastern coastal areas, forming a high-value spatial agglomeration with the trend of transferring to the central region, while the cold spots were mostly distributed in the central and western regions, forming a low-value regional agglomeration with the trend of transferring to the high value ones. The spatial distribution map of cold and hot spots for the high-quality economic development level in China is shown in Figure 4. It can be seen that the hot spots and cold spots' distribution of high-quality economic development and digital economy were very similar to each other in 2011, 2014, 2016, and 2019. The hot spots were concentrated in the eastern coastal areas, forming a high-value spatial agglomeration with the trend of transferring to the central region, while the cold spots were mostly distributed in the central and western regions, forming a low value regional agglomeration with the trend of transferring to the high value ones.

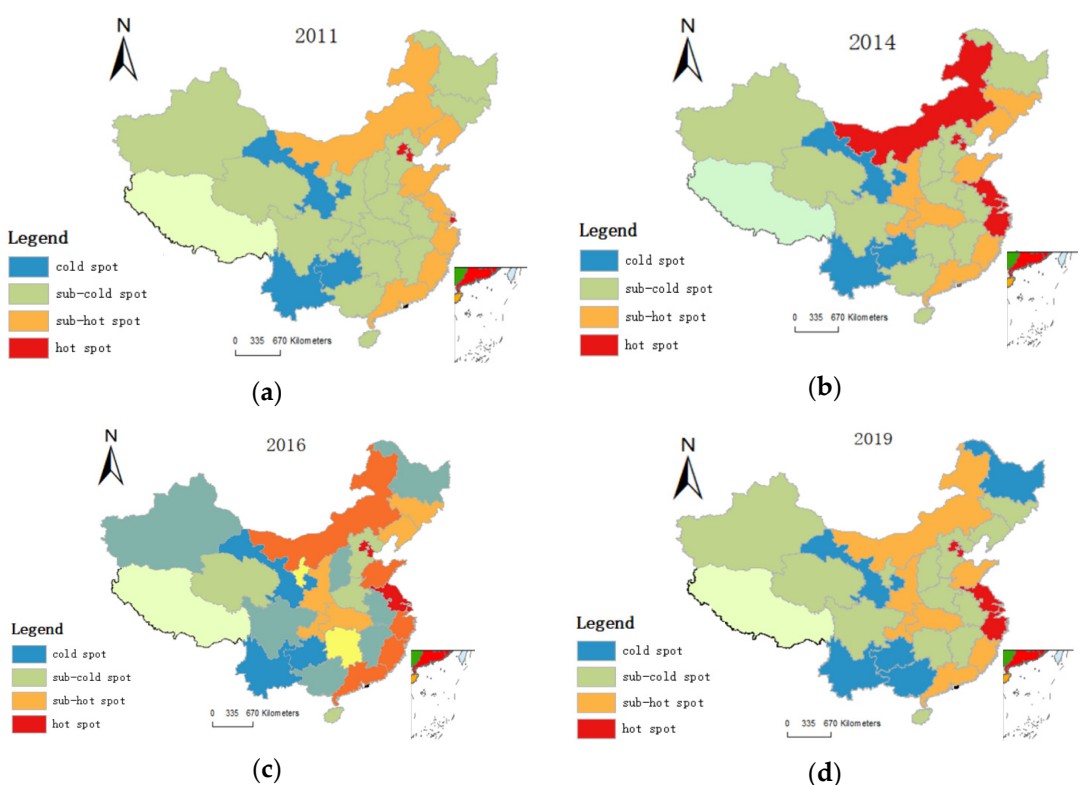

**Figure 4.** Spatial distribution map of cold and hot spots with high-quality economic development in China, which (**a**) 2011; (**b**) 2014; (**c**) 2016; (**d**) 2019.

Referring to Elhorst (2014), after the LM test, SDM model fixed effects, Hausman test, and SDM model simplification test, the SDM model with dual spatio-temporal fixed effects was selected in this section [51]. To compare the robustness of the estimations, the results of the spatial lag model (SAR) with dual spatio-temporal fixed effects were also presented. The results in Table 4 reveal that the regression coefficient of the impact of the digital economy on high-quality economic development was significantly positive under both weight matrices, which indicates that digital economy can significantly promote the development of high-quality economy. The positive effects of a digital economy in promoting high-quality economic development were verified again. In addition, as shown in Table 4, both the direct and indirect effects of a digital economy on inter-provincial high-quality economic development were significant, which supported hypothesis H5.

**Table 4.** Regression results of the spatial model of the digital economy's impact on high-quality development.

| Variables | SDM | | | SAR | | |
|---|---|---|---|---|---|---|
| | **W1** | **W2** | **W3** | **W1** | **W2** | **W3** |
| Dige | 0.103 *** | 0.079 *** | 0.095 *** | 0.101 *** | 0.108 *** | 0.114 *** |
| | (4.60) | (3.81) | (4.00) | (4.42) | (4.58) | (4.91) |
| Market | 0.009 | 0.019 *** | 0.018 ** | 0.021 ** | 0.023 *** | 0.024 *** |
| | (1.05) | (2.62) | (2.16) | (2.56) | (2.78) | (2.92) |
| FDI | −0.026 | −0.041 ** | −0.053 *** | −0.047 *** | −0.054 *** | −0.054 *** |
| | (−1.52) | (−2.49) | (−2.97) | (−2.67) | (−3.10) | (−3.10) |

**Table 4.** *Cont.*

| Variables | SDM | | | SAR | | |
|---|---|---|---|---|---|---|
| | **W1** | **W2** | **W3** | **W1** | **W2** | **W3** |
| Human | 0.054 | 0.082 ** | 0.064 | 0.014 | 0.016 | 0.009 |
| | (1.39) | (2.19) | (1.45) | (0.37) | (0.41) | (0.24) |
| Cin | 0.003 | 0.007 *** | 0.006 * | 0.004 | 0.006 ** | 0.006 ** |
| | (1.03) | (2.74) | (1.85) | (1.56) | (2.06) | (2.28) |
| Ind | 0.011 *** | 0.010 *** | 0.009 *** | 0.010 *** | 0.009 *** | 0.009 *** |
| | (9.02) | (8.97) | (6.86) | (8.14) | (7.78) | (7.83) |
| Gov | −0.019 *** | −0.017 *** | −0.017 *** | −0.018 *** | −0.018 *** | −0.018 *** |
| | (−12.07) | (−11.25) | (−9.42) | (−10.72) | (−10.43) | (−10.08) |
| _cons | −0.936 | 0.290 | 0.919 * | 1.373 *** | 0.963 *** | 0.825 *** |
| | (−1.31) | (1.20) | (1.96) | (4.26) | (4.69) | (3.73) |
| LR_Dire | 0.108 *** | 0.088 *** | 0.094 *** | 0.103 *** | 0.109 *** | 0.115 *** |
| | (4.84) | (4.21) | (3.81) | (4.35) | (4.49) | (4.81) |
| LR_Indi | −0.305 ** | −0.136 *** | −0.078 ** | −0.021 | −0.001 | 0.008 |
| | (−2.39) | (−3.89) | (−1.97) | (−1.56) | (−0.17) | (0.93) |
| LR_Total | −0.197 | −0.048 | 0.016 | 0.082 *** | 0.108 *** | 0.123 *** |
| | (−1.45) | (−1.19) | (0.32) | (3.34) | (4.20) | (4.47) |
| N | 270 | 270 | 270 | 270 | 270 | 270 |
| $R^2$-sq | 0.5799 | 0.6439 | 0.6621 | 0.6181 | 0.6615 | 0.6804 |

t-statistics or z-statistics in parentheses; *** $p < 0.01$, ** $p < 0.05$, * $p < 0.1$.

### 5.3. Further Expansion: Regional Heterogeneity

Considering the resource endowment conditions in China and the diversity at various stages of development of different regions, there is bound to be significant regional heterogeneity in the digital economy as well as in high-quality economic development. The differences presented by regional heterogeneity cannot be ignored and need to be further discussed. Therefore, the 30 provinces were divided into eastern, central, and western regions for regression estimation. The results are presented in Table 5. It is observed that, in the test for the country, eastern regions and central regions, the digital economy played a significant positive role in promoting high-quality economic development at the 1%, 5%, and 10% levels, respectively. In contrast, in the test for west regions , the role of the digital economy on high-quality economic development was not significant. In summary, after considering regional heterogeneity, the digital economy in the eastern and central regions of the country were seen to make a more significant contribution to the development of a high-quality economy. This may be because the digital economy in the eastern and central regions developed earlier and at a higher level than that in the western region. Thus, the eastern and central regions received more digital economic dividends than the western region.

**Table 5.** Regional heterogeneity test for the impact of the digital economy on high-quality development.

| Variables | The Country | Eastern Regions | Central Regions | Western Regions |
|---|---|---|---|---|
| Dige | 0.064 *** | 0.077 ** | 0.073 * | 0.032 |
| | (3.05) | (2.27) | (1.96) | (1.12) |
| TC | 0.017 *** | 0.024 * | 0.009 | 0.018 ** |
| | (2.71) | (1.91) | (0.85) | (2.34) |
| FDI | −0.061 *** | −0.253 *** | −0.063 ** | −0.014 |
| | (−4.00) | (−6.26) | (−2.61) | (−0.81) |
| Gov | −0.020 *** | −0.016 *** | −0.027 *** | −0.017 *** |
| | (−12.70) | (−4.95) | (−13.10) | (−8.46) |
| Ind | 0.007 *** | 0.009 *** | 0.003 ** | 0.007 *** |
| | (6.66) | (4.61) | (2.49) | (4.30) |
| Cin | −0.003 | −0.012 *** | 0.008 ** | −0.008 * |
| | (−1.58) | (−3.41) | (2.44) | (−1.94) |
| Human | 0.020 | −0.023 | −0.095 * | 0.108 ** |
| | (0.58) | (−0.27) | (−1.95) | (2.56) |
| Market | 0.035 *** | 0.028 *** | 0.013 | 0.023 * |
| | (4.87) | (2.67) | (1.00) | (1.99) |
| _cons | 1.422 *** | 2.933 *** | 1.341 *** | 1.258 *** |
| | (9.50) | (8.19) | (8.20) | (5.52) |
| N | 270 | 99 | 72 | 99 |
| r2_a | 0.68 | 0.68 | 0.90 | 0.76 |

t-statistics or z-statistics in parentheses; *** $p < 0.01$, ** $p < 0.05$, * $p < 0.1$.

## 6. Robustness Tests

The robustness tests were conducted from two aspects to ensure the validity of the above regression results. And the results are presented in Table 6. First, to deal with possible endogeneity problems of the model, the lagged period of the digital economy was selected as an instrumental variable for the current digital economy. The two-stage least squares of fixed-effects model was chosen for regression. Second, the control variable of the level of financial development (Fin) was added for this examination, as it had impacts on the digital economy as well as the high-quality economy through corresponding financial support. The results in the third column show that the digital economy is still significant for high-quality development after control variable Fin is added, and when Control variable(Fin) and core variable (TC) are added at the same time, the role of the digital economy in promoting the high quality of the economy is still significantly positive, as shown in the fourth columnThe results of the core explanatory variable digital economy for the two robustness tests were significantly positive at the 1% and 5% levels, indicating that the digital economy significantly contributes to high-quality economic development. These results were consistent with the results of the study presented earlier. Therefore, it can be concluded that the results of this paper are robust.

**Table 6.** Two-stage least squares regression of instrumental variables.

| Variables | Heckman Two-Stage | | Substitution Variable | |
| --- | --- | --- | --- | --- |
| | First Stage | Second Stage | Fin | Fin&TC |
| Dige | | 0.1550 *** | 0.082 *** | 0.076 ** |
| | | (0.019) | (3.02) | (2.70) |
| L.Dige | 1.0253 *** | | | |
| | (0.020) | | | |
| TC | | | | 0.014 * |
| | | | | (1.91) |
| FDI | −0.0092 | 0.0044 | −0.081 *** | −0.080 *** |
| | (0.014) | (0.014) | (−3.00) | (−2.93) |
| Gov | 0.0025 | −0.0142 *** | −0.018 *** | −0.018 *** |
| | (0.002) | (0.002) | (−6.17) | (−6.11) |
| Ind | −0.0005 | 0.0085 *** | 0.008 ** | 0.008 ** |
| | (0.001) | (0.001) | (2.68) | (2.75) |
| Cin | −0.0038 ** | 0.0276 *** | 0.002 | 0.001 |
| | (0.002) | (0.002) | (0.41) | (0.19) |
| Human | 0.0006 | −0.0429 * | 0.055 | 0.045 |
| | (0.026) | (0.025) | (0.93) | (0.74) |
| Market | 0.0196 ** | −0.0572 *** | 0.022 | 0.020 |
| | (0.010) | (0.009) | (1.31) | (1.20) |
| Fin | | | −0.029 | −0.030 |
| | | | (−1.08) | (−1.07) |
| Constant | 0.0703 | 0.1022 | 1.296 *** | 1.328 *** |
| | (0.145) | (0.141) | (3.66) | (3.65) |
| Observations | 240 | 240 | 270 | 270 |
| R-squared | 0.984 | 0.931 | 0.700 | 0.706 |

t-statistics or z-statistics in parentheses; *** $p < 0.01$, ** $p < 0.05$, * $p < 0.1$.

## 7. Conclusions and Policy Recommendations

Based on the provincial panel data of 30 provinces in China from 2011 to 2019, this paper deeply and systematically explainedthe mechanism and effect of digital economy on high-quality economic development from the aspects of direct, indirect, and heterogeneous transmission mechanisms and spatial spillover effect. The results show that: (1) digital economy could significantly promote the high-quality development of China's regional economy, which is consistent with the research results of Ge and Wu [42]; in addition, the strong diffusion characteristics of the digital economy drove the improvement of the quality of economic development in adjacent areas, which showed that the digital economy had become a strong engine for China to promote high-quality economic development in the new normal and new era. This conclusion was still valid under the robustness test. (2) As one of the important transmission paths of technological innovation and digital economy to high-quality economic development, technological innovation could indirectly promote the improvement of high-quality economic level, indicating that technological innovation and digital economy can form a driving force for high-quality economic development. This conclusion expanded the research on the transmission mechanism of digital economy on high-quality economic development. (3) There were regional differences and heterogeneity in the role of digital economy in the development of high-quality economy, and its role

showed a ladder decreasing trend. That is, in the eastern and central regions of China, its promoting effect was more significant but its driving effect on the western region had not been fully reflected. The reason could be related to resource endowment and economic location differences. At the same time, it also showed that the digital economic dividend in the western region had not been fully realized.

The above results have verified the role of digital economy in promoting high-quality economic development. However, from the perspective of regional heterogeneity, the development of China's digital economy shows an unbalanced trend, the digital divide between regions is prominent, and the role of digital economy in promoting high-quality economic development is significantly different in the east, central and western regions. Accordingly, this study puts forward the following policy implications.

(1) Strengthen digital construction and make digital economy a "sharp weapon" for high-quality economic development. On the whole, the role of regional digital economy in economic development is obvious; however, it is undeniable that there are still differences in its digital economy level and its impact on high-quality economic development. This requires that, in the construction of a digital power in the future, we should grasp the empirical law of the spatial spillover effect of the digital economy, give full play to the role of government guidance and support, strengthen the coordination of new infrastructure construction in various regions, accelerate 5G commercial application, big data model construction, and artificial intelligence application, and reasonably implement the innovation drive of the digital economy and relevant policy layout. As well, fully release its spatial contribution ability to high-quality economic development, accelerate the digital transformation process of traditional industries, and provide a good basic guarantee for the development of digital economy. On this basis, strengthen institutional innovation and reconstruct the ecosystem of digital economy development.

(2) Promote the deep integration of digital economy, technological innovation, and high-quality economic development. Government departments should promote the public data disclosure, strengthen standardized production and standardized supply of digital technology, improve data information management ability, guide the rational development of digital economy, accelerate the pace of digital industrialization and industrial digitization, and realize the breakthrough of digital economy in core technology. While vigorously developing the digital economy and improving technological innovation, we should also consolidate basic research strength, increase R & D intensity, and capital investment in core technology fields, so as to lay a foundation for technological breakthroughs in the development of digital economy. In addition, we should enhance the leading position in key core fields such as software engineering, and grasp the latest trends and technological frontiers of the global information industry. Technology transfer and independent research and development shoule be effectively promoted, to realize key technological innovation breakthroughs in the field of digital economy. And as technological innovation plays a supporting role in the deep integration of digital economy and high-quality economic development. Therefore, in the process of digital economy promoting high-quality economic development, we should fully consider the role of technological innovation between the two, and use technological innovation to drive high-quality economic development.

(3) Implement the dynamic and differentiated digital economy development strategy, and make the digital economy become the "hardware" to effectively reduce the imbalance of regional development technical support. This study found that the spatial spillover effect of digital economy can affect the quality of economic development in adjacent areas; however, there are significant regional differences in the network effect of digital economy affecting high-quality economic development, which requires that the formulation and implementation of the strategy of digital economy driving high-quality economic development should be dynamic and differentiated. Secondly, we should actively guide the formation of digital economic value and basic principles, and draw specific boundaries for strict supervision; prudently control the inflow scale of data elements, prevent malicious monopoly, and reduce unreasonable fluctuations in the industrial structure. On the basis of

consolidating the advantages of digital economy in the eastern region, moderately promote the tilt of digital elements and favorable policies to the central and western regions, guide the flow and diffusion of digital resource elements to the central and western regions, and achieve the purpose of reducing the development difference of digital economy among regions by relying on scale effect and network effect. In the eastern region with a high level of digital economy development, we should further strengthen the construction of digital economy and strengthen the innovation spillover effect of digital technology. The central and western regions should focus more on making up for weaknesses, strengthen infrastructure construction, expand the scale and coverage of digital economy, introduce large science and technology enterprises in related industries, enhance industrial ties with the eastern region, break the constraints of unbalanced spatial development of digital economy, and promote the dynamic and coordinated development of digital economy and the quality of economic development.

Although the research provides a series of empirical evidence for the promotion of high-quality development of digital economy, there are still some limitations in the paper, which will become the direction of our future research. First, the research on digital economy in this paper is concentrated at the provincial level, which is a wild and not-specific scale. Secondly, the index selection of digital economy and high-quality economic variables needs to be further improved to reflect the characteristics of regional development levels. Thirdly, the transmission mechanism between digital economy and high-quality economic development needs to be further explored and clarified to provide more feasible suggestions for the policy implementation. Finally, at present, China's digital economy and technological innovation are in the process of dynamic development, and some dynamic environmental change factors will be considered in subsequent further research. In order to cope with the changes in dynamic environment, follow-up research will also include digital twins in the scope of research.

**Author Contributions:** Conceptualization, C.D. and C.L.; methodology, C.D.; writing—original draft preparation, C.D. and C.L.; writing—review and editing, C.Z. and F.L. All authors have read and agreed to the published version of the manuscript.

**Funding:** This research was funded by the National Social Science Fund (19CGL021), the Ministry of Education of Humanities and Social Science Project (20YJC630180), and the Fundamental Research Funds for the Central Universities (B210207019).

**Institutional Review Board Statement:** Not applicable.

**Informed Consent Statement:** Informed consent was obtained from all subjects involved in the study.

**Data Availability Statement:** The data used to support the findings of this study are available from the corresponding author upon request.

**Conflicts of Interest:** The authors declare that they have no conflict of interest.

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
