# Peer review of "Digital Economy, Technological Innovation and High-Quality Economic Development: Based on Spatial Effect and Mediation Effect"

_sustainability, doi:10.3390/su14010216_

Round 1
Reviewer 1 Report
Abstract
I suggest adding introduction of your research so, It must state aims of study in first two lines.
Introduction
Add few lines about the variables you used in research i.e.
Definition of
- Digital Economy
- Technological Innovation and
- Economic Development
Literature Review
Well written
Research Design
Why authors choose to conduct such analysis i.e. significance of this analysis, add few lines in start.
Conclusion
This section shows separately
- Theoretical extensions
- Practical implications
- Limitations and future research agenda
Author Response
Reviewer #1
Thank you very much for the very insightful comments and suggestions. Base on your comment and request, we have made extensive modification on the original manuscript. In this new revision, we have tried our best to address all the concerns. A point-to-point response to your comments is shown below.
1.Abstract: I suggest adding introduction of your research so, it must state aims of study in first two lines.
Authors’ response:
Thank you very much for the meaningful comments and suggestions. According to your suggestions, we have added some contents to state the aim of this study as shown below. Please check it in the first two lines of abstract section.
"The technological innovation and high-quality economic development are inevitable requirements of sustainable development. And the digital economy has gradually become a new engine to en-hance technological innovation and the high-quality development of China’s economy. Deeply discussing the effect of digital economy on high-quality economic development and clarifying the mechanism behind it can effectively release the boosting power of digital economy to China's high-quality development, which is of great practical significance to China's sustainable economic development. "
- Introduction: Add few lines about the variables you used in research i.e. Definition of digital economy, definition of technological Innovation, definition of economic development.
Authors’ response:
Thank you for your kindly comments on this part. According to your suggestions, we have added some contents to clarify the definition of digital economy, definition of technological Innovation, definition of economic development of this study as shown below. Please check it in the measurement and description of variables section.
- On the basis of previous scholars' exposition on the concept of digital economy and international important discussions on digital economy, combined with the research content and objectives of this paper, we make the following definitions for Digital Economy: in view of the fact that the connotation and scope of digital economy have not yet formed a unified definition standard, " digital economy is understood as taking the use of digital knowledge and information as the key production factors, modern information network as the important carrier A series of economic activities with the effective use of information and communication technology as an important driving force for efficiency improvement and economic structure optimization. "
- "Technological innovation is all the activities that innovators use new technologies and inventions to change production factors and production conditions, and commercial-ize the change results. "
- " High quality economic development is in the period of economic structure transformation. China's economic growth will change from the traditional extensive economic growth to a more efficient, high-quality, economical, environmental friendly and green economic growth model, and the economic growth under this model is also high-quality economic development."
3.Research design: Why authors choose to conduct such analysis i.e. significance of this analysis, add few lines in start.
Authors’ response:
Thank you very much for the rigorous and meaningful suggestion. According to your suggestions, we have added some contents to express significance of this analysis. Please check it in the methodology and materials section.
“In previous studies, few authors have carried out systematic and in-depth theoretical analysis and quantitative research on digital economy, technological innovation and economic development. However, in the Chinese context, the relationship between the three is becoming closer and closer. It is necessary to clarify the internal logic and the-oretical basis of the three .In order to fully explore the impact of digital economy on high-quality economic development, this paper uses intermediary model and spatial model to discuss the direct, indirect and spatial spillover effects of digital economy on high-quality economic development, and explores the important role of technological innovation. The specific research methods, index selection, data sources and data pro-cessing methods are as follows.”
4.Conclusion: This section shows separately(Theoretical extensions, practical implications, limitations and future research agenda).
Authors’ response:
Thank you for your kindly comments on this part. According to your suggestions, we have made adjustments to the structure of this conclusion section, The specific modifications are shown as below.
(1) the structure has been adjusted.
(2) The limitation and future research agenda: “ Although the research provides a series of empirical evidence for the digital economy to promote high-quality development, there are still some deficiencies in the research, which will become the direction and entry point of future research. First, the research on the digital economy stays at the provincial level, does not analyze from a more detailed level, such as cities, economic belts, urban agglomerations and so on Foot; Secondly, the measurement of variables needs to be further improved in order to more accurately measure the relationship between digital economy and high-quality economic development. Thirdly, the transmission path mechanism between digital economy and high-quality economic development needs to be further excavated in order to have a deeper understanding of its mechanism; Finally, we should deeply explore the regulation mechanism of digital economy and high-quality economic development, in order to more comprehensively reveal the action mechanism of digital economy and high-quality economic development. Finally, at present, China's digital economy and technological innovation are in the process of dynamic development, and some dynamic environmental change factors will be considered in subsequent further research.”
- All the comments are carefully addressed and all the changes are highlighted by colors, which are summarized as follows.
(1)According to the reviewers’ advices, the manuscript were amended and polished to give a more clear representation. Revisions has been made on the structure and English language editing, to make it in a more formal type.
(2) In order to make the model more suitable and the result more robust, we have extended the study period, taking account of reviewers’ suggestions. And it would not affect the basic results and conclusions as we stated before.
(3) Necessary items, such as the outline of this whole paper, study limitations and descriptive statistics, has been added to make this article more complete.
(4) We have added necessary information of references in the sections of literature review, and the quantitative approaches.
Should you have any questions, please let us know and give us the opportunity to revise again. We hope that the revised version of the manuscript could be reconsidered for publication in your journal.
Reviewer 2 Report
This is an interesting article on a topical issue whose interest is indisputable. The paper is well argued at the conceptual level and its methodology is novel. The results are also well presented and relevant.
The structure of the article is also adequate from a scientific point of view. The analysis of the state of the art is sufficient to justify the relevance of the study on an international scale.
The integration of methodologies and the presentation of results are interesting. Also the conclusions and the proposal of recommendations at the policy level are significant.
At the conceptual level, the hypotheses raised are considered to be issues that have already been sufficiently justified by the scientific literature. Therefore, rather than hypotheses, they are arguments in support of the thematic line of the article. In any case, the work has merit.
It would be interesting if the article had a methodological scheme in which the set of methodologies used could be easily visualised. I think this is one of the added values of the study that can be transferred to other studies.
It would also be worthwhile to include a map showing the location of the study area so that its location at a global or continental level could be visualised.
The wording should be revised in some sections (The summary should be corrected in English and grammar): "And the digital economy...".
References to data sources are missing:
The data used in this study are 480 mainly from the China Statistical Yearbook, China Statistical Yearbook on Science and 481 Technology, and Guotaian, among which the digital economy data are from the China 482 Statistical Yearbook on Science and Technology and the China Provincial Digital Financial 483 Inclusion Index.
References to laws are missing:
Furthermore, the digital economy may also 214 have a spatial spillover effect on the development of economic quality under the com- 215 bined effect of Moore's Law and Metcalfe's Law. Based on the above, this study examines 21.
Author Response
Reviewer #2
Thank you very much for the very insightful comments and suggestions. Base on your comment and request, we have made extensive modification on the original manuscript. In this new revision, we have tried our best to address all the concerns. A point-to-point response to your comments is shown below.
1.Abstract: It would be interesting if the article had a methodological scheme in which the set of methodologies used could be easily visualized. I think this is one of the added values of the study that can be transferred to other studies..
Authors’ response:
Thank you very much for the meaningful comments and suggestions.
Actually, we have made a set of visual analysis in the article, including the spatial distribution map of digital economy and high quality, Moran scatter map and spatial agglomeration map. Considering the limited space of the article, we selected some of them instead. In the future, we will do further research on the research you proposed.
- The wording should be revised in some sections (The summary should be corrected in English and grammar): "And the digital economy...".
Authors’ response:
Thank you very much for the rigorous and meaningful suggestion. According to your suggestions, we have revised in some sections to correct in English and grammar. Please check them in the whole article.
- References to data sources are missing.
Authors’ response:
Thank you very much for the rigorous and meaningful suggestion. Base on some research, references to the date sources are shown as below. Please check it in the section of data sources and descriptive statistics section.
“The data used in this study are mainly from the China Statistical Yearbook, China Sta-tistical Yearbook on Science and Technology, and Guotaian, among which the digital economy data are from the China Statistical Yearbook on Science and Technology and EPS Global Databaseand the China Provincial Digital Financial Inclusion Index. ”
References:
[20]Song, Y.(2019). Digital economy and high-quality development from the perspective of economic development quality theory. Guizhou Social Sciences (11),102-108.
[21]Zhao, T., Zhang, Z., et al.(2020). Digital economy, entrepreneurial activity and high-quality development: empirical evidence from Chinese cities [J]. Management World (10),65-76.
[22]Wang, K.K., Wu, G.B., et al.(2020). Has the development of the digital economy improved production efficiency?[J]. Economic Research Journal (10),24-34.
[23]Jin, B.(2018). Promote the high-quality development of regional economy with innovative thinking [J]. Regional Economic Review (04),39-42.
- References to laws are missing.
Authors’ response:
Thank you very much for the meaningful comments and suggestions. According to your suggestions, we have added some references to Supplementary reference to law of this study as shown below. Please check it in the theoretical analysis and research hypotheses section.
Reference:
[21]Zhao, T., Zhang, Z., et al.(2020). Digital economy, entrepreneurial activity and high-quality development: empirical evidence from Chinese cities [J]. Management World (10),65-76.
[31]Ge Heping, Wu Fuxiang Digital economy enables high-quality economic development: theoretical mechanism and empirical evidence [J] Nanjing Social Sciences, 2021, (01): 24-33
- Furthermore, the digital economy may also have a spatial spillover effect on the development of economic quality under the combined effect of Moore's Law and Metcalfe's Law. Based on the above, this study examines 21.
Authors’ response:
Thank you for your kindly comments on this part. According to your suggestions, we have made adjustments to the content of theoretical analysis and research hypotheses, The specific modifications are shown as below.
To solve this problem, we consulted the literature of authoritative journals and inserted them into the text in the form of references. The references are as follows:
[21]Zhao, T., Zhang, Z., et al.(2020). Digital economy, entrepreneurial activity and high-quality development: empirical evidence from Chinese cities [J]. Management World (10),65-76.
[22] Ge Heping, Wu Fuxiang Digital economy enables high-quality economic development: theoretical mechanism and empirical evidence [J] Nanjing Social Sciences, 2021, (01): 24-33
- All the comments are carefully addressed and all the changes are highlighted by colors, which are summarized as follows.
(1)According to the reviewers’ advices, the manuscript were amended and polished to give a more clear representation. Revisions has been made on the structure and English language editing, to make it in a more formal type.
(2) In order to make the model more suitable and the result more robust, we have extended the study period, taking account of reviewers’ suggestions. And it would not affect the basic results and conclusions as we stated before.
(3) Necessary items, such as the outline of this whole paper, study limitations and descriptive statistics, has been added to make this article more complete.
(4) We have added necessary information of references in the sections of literature review, and the quantitative approaches.
Should you have any questions, please let us know and give us the opportunity to revise again. We hope that the revised version of the manuscript could be reconsidered for publication in your journal.
Reviewer 3 Report
First of all, I want to thank you for the effort to write this manuscript but I have some important considerations that in my point of view are of high importance:
Comments on title, abstract, references
The title matches the abstract well. It conveys the main idea of the article and also is somewhat interesting.
- 2. The abstract is well designed regarding the aim, and brief method, but a lack of the key results.
3. The references are relevant and somewhat recent. To the best of my knowledge, they correctly cited and no major references are missing.
Comments on introduction/background
1. The study context is acceptable. However, it could be better by developing the importance for the subject of interest.
2. The background in the introduction could be improved by adding more related and recent reference.
3. After a search in Google Scholar, I convinced that the most recent relevant studies are included.
4. The research question is outlined clearly.
5. The contribution adds to the literature is interested.
6. The Hypotheses in not justified well and need more references and justifications.
Comments on methodology
1. The variables are measured appropriately.
2. It would better to more clarifying the data source… As authors indicated that, (this study selects the 475 panel data of 30 provinces in China from 2011 to 2018). However, we are currently in the ending of 2021.
Comments on discussion and descriptive statistics
1. This section appropriately begins with a brief statement of the key findings. The aims are answered and the data truly supports the conclusions.
2. The use of supporting evidence in not perfect. It would be better to link between the findings and previous studies (existing knowledge) and their comparison.
Comments on conclusions and policy recommendations
- The Implications also need to be improved. The paper lacks of clear Implications to the theory development and management.
- The limitations and the suggestions of future research in the area need to be more addressed.
I hope that all these questions and comments can help to improve your manuscript.
Author Response
Reviewer #3
Thank you very much for the very insightful comments and suggestions. Base on your comment and request, we have made extensive modification on the original manuscript. In this new revision, we have tried our best to address all the concerns. A point-to-point response to your comments is shown below.
- abstract: The abstract is well designed regarding the aim, and brief method, but a lack of the key results.
Authors’ response:
Thank you very much for the meaningful comments and suggestions. According to your suggestions, we have added some contents to state the key results of this study as shown below. Please check it in the abstract section.
(1) “The research results show that the overall level of digital economy and high-quality development is not high, there are high-high agglomeration and low-low agglomeration, and the phenomenon of spatial path dependence and spatial locking is obvious.”
(2) “digital economy can promote the improvement of high-quality economic development level, and the spatial spillover effect is significant, and the robustness test results such as two-stage least squares regression and substitution variables are consistent."
In addition, the role of digital economy in promoting the high-quality economic development of the eastern, central and western regions is gradually weakened. Technological innovation is an important transmission path of digital economy to high-quality economic development.”
- introduction/background:The study context is acceptable. However, it could be better by developing the importance for the subject of interest.
Authors’ response:
Thank you very much for the rigorous and meaningful suggestion. According to your suggestions, we have added some contents to develop the importance for the subject of interest. Please check it in the introduction section.
- introduction/background: The background in the introduction could be improved by adding more related and recent reference.
Authors’ response:
Thank you very much for the rigorous and meaningful suggestion. According to your suggestions, we have added some contents to improve the background by adding more related and recent reference in the introduction for the subject of interest. Please check it in the introduction section.
In order to more clearly explain the existing research status, theoretical basis and research context in the research field of this paper, closely follow the theme of the article, and add a number of more representative and explanatory relevant literature
For example:
- Guo, Liang . The impact mechanism of the digital economy on China's total factor productivity: an uplifting effect or a restraining effect? [J]. Southern Economy, 2021(10): 9-27.
- Ding, Z.F.(2020). Research on the mechanism of digital economy driving high-quality economic development: a theoretical analysis framework [J]. Modern Economic Research (01),85-92. doi:10.13891/j.cnki.mer.2020.01.011.
- Li Zongxian, Yang Qianfan. How does the digital economy affect the high-quality development of China's economy? [J]. Modern Economic Research, 2021(07): 10-19.
- Lu Yuxiu, Fang Xingming, Zhang Jianan Digital economy, spatial spillover and high-quality development of urban economy [J] Economic longitude and latitude, 2021,38 (06): 21-31.
4.introduction/background: The Hypotheses in not justified well and need more references and justifications.
Authors’ response:
Thank you very much for the rigorous and meaningful suggestion. According to your suggestions, we have added more references and justifications to state the hypotheses better. Please check it in the Theoretical analysis and research hypotheses.
(1) Direct impact mechanism of digital economy on high economic quality
[18]Hong, Y.X.(2018). Cultivating new momentum: an upgraded version of supply-side structural reform [J]. Economic Science (03),5-13.
[12]Zhang, T., Jiang ,F.X., et al.(2021). Can digital economy become a new kinetic energy to promote the high-quality development of China's economy?. Inquiry into Economic Issues (01),25-39. doi:CNKI:SUN:JJWS.0.2021-01-004.
[13]Guo, J.T., Luo, P.L.(2016). Does the Internet promote China's total factor productivity?. Management World (10),34-49.
[9]Czernich, N.,T. Falck,and L. Woessmann. Broadband Infrastructure and Economic Growth [J]. Economic Journal,2011,121(552):505-532.
[14]Li, X. Z., Yang, Q.F. (2021). How does digital economy affect the high-quality development of Chinese economy?. Modern Economic Research(07),10-19.
[15]Yang, H.M., Jiang, L.(2021). Digital Economy, Spatial Effect and Total Factor Productivity. Statistical Research (04),3-15.
etc
(2) Indirect impact mechanism of digital economy on high-quality economic development
[2]Guo, Liang . The impact mechanism of the digital economy on China's total factor productivity: an uplifting effect or a restraining effect? [J]. Southern Economy, 2021(10): 9-27.
[14]Zhang, T., Jiang ,F.X., et al.(2021). Can digital economy become a new kinetic energy to promote the high-quality development of China's economy?. Inquiry into Economic Issues (01),25-39. doi:CNKI:SUN:JJWS.0.2021-01-004.
[19]Wang, J.(2019). Digital economy drives high-quality economic development: factor allocation and strategic choice [J]. Social Sciences in Ningxia (05),88-94.
[20]Song, Y.(2019). Digital economy and high-quality development from the perspective of economic development quality theory. Guizhou Social Sciences (11),102-108.
[32]Zhang Y, Dong C, Luan J. Research on the mechanism of digital economy promoting high-quality economic development -- evidence based on Provincial Panel Data [J] Journal of Jinan University (SOCIAL SCIENCE EDITION), 2021,31 (05): 99-115 +
(3) Spatial spillover mechanism of digital economy to high-quality economic development.
[34]Bian, Z.Q.(2014). Research on Spillover Effect and Mechanism of Network Infrastructure [J]. Journal of Shanxi University of Finance and Economics (09),72-80. doi:10.13781/j.cnki.1007-9556.2014.09.006.
[35]Li, T.Z., Wang, W.(2018). Comparative Research on the Spatial Spillover Effects of Network Infrastructure. East China Economic Management (12),5-12. doi:10.19629/j.cnki.34-1014/f.180105029.
[35]Lin, Juan, Yu, et al. Sustainability, Vol. 9, Pages 946: Internet Access, Spillover and Regional Development in China. 2017.
[36]Zhang, J.Y., Guo, K.G., et al.(2019). E-commerce development, spatial spillover and economic growth: based on empirical evidence from prefecture-level cities in China [J]. Finance & Economics (03),105-118. doi:CNKI:SUN:CJKX.0.2019-03-010.
[37]Li, X.Z., Wang, H.(2020). A Comparative Study on the Regional Difference of the Influence of the Internet on my country's Economic Development [J]. China Soft Science (12),22-32. doi:CNKI:SUN:ZGRK.0.2020-12-003.
[38]Han, C.G., Zhang, L.(2019). Does the Internet Improve China’s Resource Misallocation——Based on the Test of the Dynamic Space Dubin Model and the Threshold Model (12),43-55. doi:CNKI:SUN:JJWS.0.2019-12-004.
Etc
- methodology: It would better to more clarifying the data source… As authors indicated that, (this study selects the 475 panel data of 30 provinces in China from 2011 to 2018). However, we are currently in the ending of 2021.
Authors’ response:
Thank you very much for the meaningful comments and suggestions. According to your suggestions, We have updated the data of the paper and clarified the data source, The specific modifications are shown as below.
(1)“The data used in this study are mainly from the China Statistical Yearbook, China Sta-tistical Yearbook on Science and Technology, and Guotaian, among which the digital economy data are from the China Statistical Yearbook on Science and Technology and EPS Global Databaseand the China Provincial Digital Financial Inclusion Index. ”
References:
[20]Song, Y.(2019). Digital economy and high-quality development from the perspective of economic development quality theory. Guizhou Social Sciences (11),102-108.
[21]Zhao, T., Zhang, Z., et al.(2020). Digital economy, entrepreneurial activity and high-quality development: empirical evidence from Chinese cities [J]. Management World (10),65-76.
[22]Wang, K.K., Wu, G.B., et al.(2020). Has the development of the digital economy improved production efficiency?[J]. Economic Research Journal (10),24-34.
[23Jin, B.(2018). Promote the high-quality development of regional economy with innovative thinking [J].Regional Economic Review (04),39-42.
(2)The paper have extended the study period to 2019. It can be checked in the website of “National data (stats.gov.cn)”China Economic and social development statistical database - Yearbook Navigation (cnki.net).
- discussion and descriptive statistic: The use of supporting evidence in not perfect. It would be better to link between the findings and previous studies (existing knowledge) and their comparison.
Authors’ response:
Thank you very much for the rigorous and meaningful suggestion. According to your suggestions, we have added some supporting evidences to this paper. Please check it in the conclusions and policy recommendations.
For example:
Digital economy can significantly promote the high-quality development of China's regional economy, which is consistent with the results of Ge and Wu.
[1] Ge, Wu .Digital economy enables high-quality economic development: theoretical mechanism and empirical evidence [J] Nanjing Social Sciences, 2021 (01): 24-33.
- conclusions and policy recommendations: The Implications also need to be improved. The paper lacks of clear Implications to the theory development and management.
Authors’ response:
Thank you for your kindly comments on this part. According to your suggestions, we have made adjustments to conclusions and policy recommendations of this conclusion section, The specific modifications are shown in the inconclusions and policy recommendations.
- conclusions and policy recommendations: The limitations and the suggestions of future research in the area need to be more
addressed.
Authors’ response:
Thank you for your kindly comments on this part. According to your suggestions, we have added some contents to express the limitations and the suggestions of future research in the area. Please check it in the conclusions and policy recommendations.
“ Although the research provides a series of empirical evidence for the digital economy to promote high-quality development, there are still some deficiencies in the research, which will become the direction and entry point of future research. First, the research on the digital economy stays at the provincial level, does not analyze from a more detailed level, such as cities, economic belts, urban agglomerations and so on Foot; Secondly, the measurement of variables needs to be further improved in order to more accurately measure the relationship between digital economy and high-quality economic development. Thirdly, the transmission path mechanism between digital economy and high-quality economic development needs to be further excavated in order to have a deeper understanding of its mechanism; Finally, we should deeply explore the regulation mechanism of digital economy and high-quality economic development, in order to more comprehensively reveal the action mechanism of digital economy and high-quality economic development. Finally, at present, China's digital economy and technological innovation are in the process of dynamic development, and some dynamic environmental change factors will be considered in subsequent further research.”
- All the comments are carefully addressed and all the changes are highlighted by colors, which are summarized as follows.
(1)According to the reviewers’ advices, the manuscript were amended and polished to give a more clear representation. Revisions has been made on the structure and English language editing, to make it in a more formal type.
(2) In order to make the model more suitable and the result more robust, we have extended the study period, taking account of reviewers’ suggestions. And it would not affect the basic results and conclusions as we stated before.
(3) Necessary items, such as the outline of this whole paper, study limitations and descriptive statistics, has been added to make this article more complete.
(4) We have added necessary information of references in the sections of literature review, and the quantitative approaches.
Should you have any questions, please let us know and give us the opportunity to revise again. We hope that the revised version of the manuscript could be reconsidered for publication in your journal.
Reviewer 4 Report
Dear Authors, thank You for so interesting research!
- General concept comments.
The article is written on the relevant topic and is well structured as well as logically proved. The chosen topic is very interesting from both the theoretical and practical points of view. No doubt, the digital economy has a great impact on economic growth.
However, I'd recommend making some improvements to the structure of the article:
- Please kindly shorten the section 1 Introduction setting aside the explanation with large number of references and help the reader understanding the following important issues: the research gap which the authors could disclose in the article; the relevance of the topic; the research question and the aim of the article. All those elements could be described on one page (the last three paragraphs perfectly reflects the mentioned issues);
- Please add the Discussion section containing the limitations of the model and the issues for future research.
The lines 191-192 contains the following sentence: "In summary, given that the relationship between the digital economy and high-quality development has been proposed relatively recently." Please kindly make it better for understanding. I'm not sure that the concept digital economy has been implemented separately from economic growth. On the contrary, I'd recommend to make the proposition about full coherence between the digital economy and technological innovations.
- Scientific Novelty.
Please kindly describe the scientific contribution to the theory. For example, the researchers could consider stronger argument about the possibility of digital coordination of economic development all over the national economy including the eastern and central regions - instead of weaker argument and more practical provision in the lines 672-674: "Additionally, the spatial spillover effect of the digital economy in high-quality economic development can help coordinate the economic development pattern between the eastern and central regions and solve the problem of uneven regional economic development.".
- General questions.
There is dome unclear issues regarding the disclosing such definitions as digital twins and digital logistics being implemented in the era of digitalization. Otherwise the Article looks like as paper dedicated mostly to innovations. Also it could be recommended to add to the literature review some references regarding the wider understanding of the Concept of digital twins and digital logistics :
Barykin, S.Y., Kapustina, I.V., Sergeev, S.M., Kalinina, O.V., Vilken, V.V., de la Poza, E., Putikhin, Y.Y., Volkova, L.V. Developing the physical distribution digital twin model within the trade network (2021) Academy of Strategic Management Journal, 20 (SpecialIssue2), pp. 1-18. https://www.scopus.com/inward/record.uri?eid=2-s2.0-85106875305&partnerID=40&md5=db6f042b3d2623c43c8b21e13f470776
Barykin, S.Y., Bochkarev, A.A., Sergeev, S.M., Baranova, T.A., Mokhorov, D.A., Kobicheva, A.M. A Methodology Of Bringing Perspective Innovation Products To Market (2021) Academy of Strategic Management Journal, 20 (SpecialIssue2), pp. 1-19. https://www.scopus.com/inward/record.uri?eid=2-s2.0-85107807314&partnerID=40&md5=7ed0efc3e4200609113740a030902141
- Ethics statements and data availability statements.
In view of practical significance of the developed approach with the data used, the authors could be asked about the correct understanding and following both the recommended guidelines of the Committee on Publication Ethics (https://publicationethics.org/) and Institutional Review Board Written Procedures: Guidance for Institutions and IRBs (2018).
Author Response
Reviewer #4
Thank you very much for the very insightful comments and suggestions. Base on your comment and request, we have made extensive modification on the original manuscript. In this new revision, we have tried our best to address all the concerns. A point-to-point response to your comments is shown below.
1.General concept comments.
(1)Please kindly shorten the section 1 Introduction setting aside the explanation with large number of references and help the reader understanding the following important issues: the research gap which the authors could disclose in the article; the relevance of the topic; the research question and the aim of the article. All those elements could be described on one page (the last three paragraphs perfectly reflects the mentioned issues)
Authors’ response:
Thank you for your kindly comments on this part
According to your suggestion, We edited the introduction to make this part look more concise, making the content of the paper closely related to the research topic. In addition, we also adjusted the structure of the article. Please check it in the introduction section.
“We condensed and compressed the introduction to make the language more concise and concise on the basis of ensuring the complete expression of meaning.”
(2)Please add the Discussion section containing the limitations of the model and the issues for future research.
Authors’ response:
Thank you for your kindly comments on this part. According to your suggestions, we have added the discussion section in this conclusion section, The specific modifications are shown as below.
“Although the research provides a series of empirical evidence for the digital economy to promote high-quality development, there are still some deficiencies in the research, which will become the direction and entry point of future research. First, the research on the digital economy stays at the provincial level, does not analyze from a more detailed level, such as cities, economic belts, urban agglomerations and so on Foot; Secondly, the measurement of variables needs to be further improved in order to more accurately measure the relationship between digital economy and high-quality economic development. Thirdly, the transmission path mechanism between digital economy and high-quality economic development needs to be further excavated in order to have a deeper understanding of its mechanism; Finally, we should deeply explore the regulation mechanism of digital economy and high-quality economic development, in order to more comprehensively reveal the action mechanism of digital economy and high-quality economic development. Finally, at present, China's digital economy and technological innovation are in the process of dynamic development, and some dynamic environmental change factors will be considered in subsequent further research. "As an explanation of this problem.”
(3)The lines 191-192 contains the following sentence: "In summary, given that the relationship between the digital economy and high-quality development has been proposed relatively recently." Please kindly make it better for understanding. I'm not sure that the concept digital economy has been implemented separately from economic growth. On the contrary, I'd recommend to make the proposition about full coherence between the digital economy and technological innovations.
Authors’ response:
Thank you very much for the meaningful comments and suggestions. According to your suggestions, we have added some contents to state the aim of this passage as shown below. Please check it in the literature review section.
“Considering the research of many other scholars, and closely follow the research theme and research background of this paper, We explain the relationship between digital economy, technological innovation and economic development as follows: relying on the strong diffusion of digital economy based on digital technology and new infrastructure, give full play to the multiplier effect of data elements on the efficiency of other production factors, help release the new vitality of economic development, help improve total factor productivity and promote high-quality economic development. Digital economy can actively promote technological innovation and product innovation, so as to significantly improve the regional innovation ability. "
2.Scientific Novelty.
Please kindly describe the scientific contribution to the theory. For example, the researchers could consider stronger argument about the possibility of digital coordination of economic development all over the national economy including the eastern and central regions - instead of weaker argument and more practical provision in the lines 672-674: "Additionally, the spatial spillover effect of the digital economy in high-quality economic development can help coordinate the economic development pattern between the eastern and central regions and solve the problem of uneven regional economic development."
Authors’ response:
Thank you for your kindly comments on this part. According to your suggestions, we have added some contents to clarify the content. Please check it in the paper.
“In previous studies, few authors have carried out systematic and in-depth theoretical analysis and quantitative research on digital economy, technological innovation and economic development, but in the Chinese context, the relationship between the three is becoming closer and closer. This paper expounds the internal logic and theoretical basis of the three. Considering that the economic level of the central and western regions lags behind that of the eastern region, but there is a large room for improvement, speed up the construction of digital economy, make full use of the innovation spillover dividends obtained from the development of digital economy, and grasp the opportunity of "overtaking in corners". Based on this, the state should grasp the empirical law of spatial spillover effect of digital economy, strengthen the coordination of new infrastructure construction in various regions, reasonably implement the innovation drive of digital economy and relevant policy layout, and fully release its spatial contribution to high-quality economic development, that is, while driving high-quality economic development, we should also take into account the narrowing of the gap between regions.” As an explanation of this problem.
- General questions.
There is dome unclear issues regarding the disclosing such definitions as digital twins and digital logistics being implemented in the era of digitalization. Otherwise the Article looks like as paper dedicated mostly to innovations. Also it could be recommended to add to the literature review some references regarding the wider understanding of the Concept of digital twins and digital logistics :
Barykin, S.Y., Kapustina, I.V., Sergeev, S.M., Kalinina, O.V., Vilken, V.V., de la Poza, E., Putikhin, Y.Y., Volkova, L.V. Developing the physical distribution digital twin model within the trade network (2021) Academy of Strategic Management Journal, 20 (SpecialIssue2), pp. 1-18. https://www.scopus.com/inward/record.uri?eid=2-s2.0-85106875305&partnerID=40&md5=db6f042b3d2623c43c8b21e13f470776
Barykin, S.Y., Bochkarev, A.A., Sergeev, S.M., Baranova, T.A., Mokhorov, D.A., Kobicheva, A.M. A Methodology Of Bringing Perspective Innovation Products To Market (2021) Academy of Strategic Management Journal, 20 (SpecialIssue2), pp. 1-19.https://www.scopus.com/inward/record.uri?eid=2-s2.0-85107807314&partnerID=40&md5=7ed0efc3e4200609113740a030902141
Authors’ response:
Thank you very much for the meaningful comments and suggestions. According to your suggestions, we have added some reference to enrich the content of literature review as shown below. Please check it in the literature review section and conclusion.
“In order to more clearly explain the existing research status, theoretical basis and research context in the research field of this paper, closely follow the theme of the article, and add a number of more representative and explanatory relevant literature. In addition, we have also introduced two high-quality articles you recommended into our articles to enrich our research.”
4.Ethics statements and data availability statements.
In view of practical significance of the developed approach with the data used, the authors could be asked about the correct understanding and following both the recommended guidelines of the Committee on Publication Ethics (https://publicationethics.org/) and Institutional Review Board Written Procedures: Guidance for Institutions and IRBs (2018).
Authors’ response:
Thank you for your kindly comments on this part. According to your suggestions, we have clarified the correct understanding and following both the recommended guidelines of the Committee on Publication Ethics (https://publicationethics.org/) and Institutional Review Board Written Procedures: Guidance for Institutions and IRBs (2018). IRBs helps us recognize some of our shortcomings and will continue to follow IRBs in academic research in the future.
- All the comments are carefully addressed and all the changes are highlighted by colors, which are summarized as follows.
(1)According to the reviewers’ advices, the manuscript were amended and polished to give a more clear representation. Revisions has been made on the structure and English language editing, to make it in a more formal type.
(2) In order to make the model more suitable and the result more robust, we have extended the study period, taking account of reviewers’ suggestions. And it would not affect the basic results and conclusions as we stated before.
(3) Necessary items, such as the outline of this whole paper, study limitations and descriptive statistics, has been added to make this article more complete.
(4) We have added necessary information of references in the sections of literature review, and the quantitative approaches.
Should you have any questions, please let us know and give us the opportunity to revise again. We hope that the revised version of the manuscript could be reconsidered for publication in your journal.
Reviewer 5 Report
This is a paper that looks at up-to-date topics for the Chinese economy and it uses recent bibliographical documentation.
Some comments and suggestions for revision are:
- the concepts of high quality economic development and high quality economy (as main concepts of the paper) need to be defined upfront before being discussed/commented/etc.
- section 2 of literature review it is not and it cannot be a systematic literature review and it should not be presented as such.
- the authors claim (p. 4, raws 199-200) that the study enriches theoretical research, but the study does not actually increases theoretical research (there is no theoretical development) and should not be presented in this way.
- under the formulation of research hypotheses, there is not enough theoretical grounding, there are not not enough bibliographical references used. The justification of hypotheses needs to be done based on theory and more references are to be included for each hypothesis. Only H5 is better rooted in theory, while H2, H3 and H4 have no theoretical justification at all.
- under the measurement and description of variables, not clear how the real GDP/ capita depict the high quality of economic development. Better justification is needed here.
- here is mistake/confusion between explained and explanatory variables (p. 9, raw 428)
- in terms of sources and statistical data, it needs to be clearly specified what were the exact data sources used for each indicator. An overall listing of the different sources for the whole bunch of variables it is not enough.
- the methodology is not fully explained. Variables are not all explained in terms of specific indicators used to measure them and the data sources used for each of them.
- legends for the figures are not visible
- there are still editing mistakes in the paper.
Author Response
Reviewer #5
Thank you very much for the very insightful comments and suggestions. Base on your comment and request, we have made extensive modification on the original manuscript. In this new revision, we have tried our best to address all the concerns. A point-to-point response to your comments is shown below.
1.the concepts of high quality economic development and high quality economy (as main concepts of the paper) need to be defined upfront before being discussed/commented/etc.
Authors’ response:
Thank you for your kindly comments on this part. According to your suggestions, we have added some contents to clarify the definition of digital economy, definition of technological Innovation, definition of economic development of this study as shown below. Please check it in the measurement and description of variables section.
- On the basis of previous scholars' exposition on the concept of digital economy and international important discussions on digital economy, combined with the research content and objectives of this paper, we make the following definitions for Digital Economy: in view of the fact that the connotation and scope of digital economy have not yet formed a unified definition standard, " digital economy is understood as taking the use of digital knowledge and information as the key production factors, modern information network as the important carrier A series of economic activities with the effective use of information and communication technology as an important driving force for efficiency improvement and economic structure optimization. "
- "Technological innovation is all the activities that innovators use new technologies and inventions to change production factors and production conditions, and commercial-ize the change results. "
- " High quality economic development is in the period of economic structure transformation. China's economic growth will change from the traditional extensive economic growth to a more efficient, high-quality, economical, environmental friendly and green economic growth model, and the economic growth under this model is also high-quality economic development."
“Digital economy refers to a series of economic activities with the use of digital knowledge and information as key production factors, modern information network as an important carrier, and the effective use of information and communication technology as an important driving force for efficiency improvement and economic structure optimization. As a product of the development of information and communication technology, digital economy is a powerful driving force for the country to release the efficiency of digital innovation, improve the rate of technological innovation and promote the upgrading of industrial structure, so as to speed up the economic construction from "quantitative growth" to "quality growth". Compared with the traditional extensive economic growth, the economic development promoted by digital economy improves the technical content of the total economy, promotes the transformation and upgrading of industrial structure, and gives birth to the model of coordinated development of environment-friendly economy and environment, that is, high-quality economic development.”
2.section 2 of literature review it is not and it cannot be a systematic literature review and it should not be presented as such.
Authors’ response:
Thank you for your insightful comments and suggestions. Thank you very much for the meaningful comments and suggestions. According to your suggestions, we have modified literature review of this study as shown below. Please check it in the literature review section.
“We have re sorted the literature review according to your problems and learned from the writing mode of authoritative journals, which has greatly improved the writing of our literature review and the quality of our papers. The revised literature review closely follows the theme to ensure not only systematicness, but also refinement and logical coherence.”
3.the authors claim (p. 4, raws 199-200) that the study enriches theoretical research, but the study does not actually increase theoretical research (there is no theoretical development) and should not be presented in this way.
Authors’ response:
Thank you very much for the rigorous and meaningful suggestion. According to your suggestions, We have revised some contents and sought the help of professional polishing institutions to ensure that our meaning can be accurately expressed. Please check it in the Literature review.
4.under the formulation of research hypotheses, there is not enough theoretical grounding, there are not not enough bibliographical references used. The justification of hypotheses needs to be done based on theory and more references are to be included for each hypothesis. Only H5 is better rooted in theory, while H2, H3 and H4 have no theoretical justification at all.
Authors’ response:
Thank you very much for the rigorous and meaningful suggestion. According to your suggestions, we have added more references and justifications to state the hypotheses better. Please check it in the Theoretical analysis and research hypotheses.
(1) Direct impact mechanism of digital economy on high economic quality
[18]Hong, Y.X.(2018). Cultivating new momentum: an upgraded version of supply-side structural reform [J]. Economic Science (03),5-13.
[12]Zhang, T., Jiang ,F.X., et al.(2021). Can digital economy become a new kinetic energy to promote the high-quality development of China's economy?. Inquiry into Economic Issues (01),25-39. doi:CNKI:SUN:JJWS.0.2021-01-004.
[13]Guo, J.T., Luo, P.L.(2016). Does the Internet promote China's total factor productivity?. Management World (10),34-49.
[9]Czernich, N.,T. Falck,and L. Woessmann. Broadband Infrastructure and Economic Growth [J]. Economic Journal,2011,121(552):505-532.
[14]Li, X. Z., Yang, Q.F. (2021). How does digital economy affect the high-quality development of Chinese economy?. Modern Economic Research(07),10-19.
[15]Yang, H.M., Jiang, L.(2021). Digital Economy, Spatial Effect and Total Factor Productivity. Statistical Research (04),3-15.
etc
For example:
(2) Indirect impact mechanism of digital economy on high-quality economic development
[2]Guo, Liang . The impact mechanism of the digital economy on China's total factor productivity: an uplifting effect or a restraining effect? [J]. Southern Economy, 2021(10): 9-27.
[14]Zhang, T., Jiang ,F.X., et al.(2021). Can digital economy become a new kinetic energy to promote the high-quality development of China's economy?. Inquiry into Economic Issues (01),25-39. doi:CNKI:SUN:JJWS.0.2021-01-004.
[19]Wang, J.(2019). Digital economy drives high-quality economic development: factor allocation and strategic choice [J]. Social Sciences in Ningxia (05),88-94.
[20]Song, Y.(2019). Digital economy and high-quality development from the perspective of economic development quality theory. Guizhou Social Sciences (11),102-108.
[32]Zhang Y, Dong C, Luan J. Research on the mechanism of digital economy promoting high-quality economic development -- evidence based on Provincial Panel Data [J] Journal of Jinan University (SOCIAL SCIENCE EDITION), 2021,31 (05): 99-115 +
5.under the measurement and description of variables, not clear how the real GDP/ capita depict the high quality of economic development. Better justification is needed here.
Authors’ response:
Thank you very much for the rigorous and meaningful suggestion. According to your suggestions, We refer to authoritative journals to prove that our selected alternative variables can still support our research. Please check it in the measurement and description of variables.
For example:
[42]Ge, H.P., Wu, F.X.(2021). Digital economy empowers high-quality economic development: theoretical mechanism and empirical evidence [J]. Nanjing Journal of Social Sciences (01),24-33. doi:10.15937/j.cnki.issn1001-8263.2021.01.003.
[43]Guo, F., Chen, K.(2021). The Influence of the Development of the Internet on the Urban Environmental Quality from the Spatial Perspective——Based on the Spatial Dubin Model and the Mediating Effect Model [J]. Inquiry into Economic Issues (01),104-112.
[52] Chen Shiyi, Chen Dengke Haze pollution, government governance and high-quality economic development [J] Economic research, 2018,53 (02): 20-34
[53] Lu , Fang , Zhang. Digital economy, spatial spillover and high-quality development of urban economy [J] Economic longitude and latitude, 2021,38 (06): 21-31.
ect
6.here is mistake/confusion between explained and explanatory variables (p. 9, raw 428)
Authors’ response:
Thank you very much for the meaningful comments and suggestions. According to your suggestions, We have corrected the spelling mistakes, sought the help of professional institutions, and re polished, translated and proofread the full text, so as to try our best to make our research more professional. Please check it in the Measurement and description of variables.
7.in terms of sources and statistical data, it needs to be clearly specified what were the exact data sources used for each indicator. An overall listing of the different sources for the whole bunch of variables it is not enough.
Authors’ response:
Thank you very much for the meaningful comments and suggestions. According to your suggestions, We have updated the data of the paper and more clarified the data source, The specific modifications are shown as below.
“The data used in this study are mainly from the China Statistical Yearbook, China Sta-tistical Yearbook on Science and Technology, and Guotaian, among which the digital economy data are from the China Statistical Yearbook on Science and Technology and EPS Global Databaseand the China Provincial Digital Financial Inclusion Index. ”
References:
[20]Song, Y.(2019). Digital economy and high-quality development from the perspective of economic development quality theory. Guizhou Social Sciences (11),102-108.
[21]Zhao, T., Zhang, Z., et al.(2020). Digital economy, entrepreneurial activity and high-quality development: empirical evidence from Chinese cities [J]. Management World (10),65-76.
[22]Wang, K.K., Wu, G.B., et al.(2020). Has the development of the digital economy improved production efficiency?[J]. Economic Research Journal (10),24-34.
[23Jin, B.(2018). Promote the high-quality development of regional economy with innovative thinking [J].Regional Economic Review (04),39-42.8.
8.the methodology is not fully explained. Variables are not all explained in terms of specific indicators used to measure them and the data sources used for each of them.
Authors’ response:
Thank you for your kindly comments on this part. According to your suggestions, we have added some contents to clarify the definition of digital economy, definition of technological Innovation, definition of economic development of this study as shown below. Please check it in the measurement and description of variables section.
(1)On the basis of previous scholars' exposition on the concept of digital economy and international important discussions on digital economy, combined with the research content and objectives of this paper, we make the following definitions for Digital Economy: in view of the fact that the connotation and scope of digital economy have not yet formed a unified definition standard, " digital economy is understood as taking the use of digital knowledge and information as the key production factors, modern information network as the important carrier A series of economic activities with the effective use of information and communication technology as an important driving force for efficiency improvement and economic structure optimization. "
(2)"Technological innovation is all the activities that innovators use new technologies and inventions to change production factors and production conditions, and commercial-ize the change results. "
(3)" High quality economic development is in the period of economic structure transformation. China's economic growth will change from the traditional extensive economic growth to a more efficient, high-quality, economical, environmental friendly and green economic growth model, and the economic growth under this model is also high-quality economic development."
[20]Song, Y.(2019). Digital economy and high-quality development from the perspective of economic development quality theory. Guizhou Social Sciences (11),102-108.
[21]Zhao, T., Zhang, Z., et al.(2020). Digital economy, entrepreneurial activity and high-quality development: empirical evidence from Chinese cities [J]. Management World (10),65-76.
[22]Wang, K.K., Wu, G.B., et al.(2020). Has the development of the digital economy improved production efficiency?[J]. Economic Research Journal (10),24-34.
[23]Jin, B.(2018). Promote the high-quality development of regional economy with innovative thinking [J].Regional Economic Review (04),39-42.
9.legends for the figures are not visible
Authors’ response:
Thank you for your kindly comments on this part. According to your suggestions, we have enlarged the visual legend in the text to the same scale to ensure that it can be clearly seen by readers. Please check it in the paper.
10.there are still editing mistakes in the paper.
Authors’ response:
Thank you very much for the rigorous and meaningful suggestion. According to your suggestions, We have revised some contents and sought the help of professional polishing institutions to ensure that our meaning can be accurately expressed. Please check it in the paper.
- All the comments are carefully addressed and all the changes are highlighted by colors, which are summarized as follows.
(1)According to the reviewers’ advices, the manuscript were amended and polished to give a more clear representation. Revisions has been made on the structure and English language editing, to make it in a more formal type.
(2) In order to make the model more suitable and the result more robust, we have extended the study period, taking account of reviewers’ suggestions. And it would not affect the basic results and conclusions as we stated before.
(3) Necessary items, such as the outline of this whole paper, study limitations and descriptive statistics, has been added to make this article more complete.
(4) We have added necessary information of references in the sections of literature review, and the quantitative approaches.
Should you have any questions, please let us know and give us the opportunity to revise again. We hope that the revised version of the manuscript could be reconsidered for publication in your journal.
Round 2
Reviewer 3 Report
Thank you for sending me the revised version of the manuscript. Overall, I had the impression that the authors team incorporated my concerns in the manuscript.
Reviewer 5 Report
The authors put an effort to include the reviewers' suggestions and they managed to do so.
There are still some editing aspects to be revised: page 8 raws 352-356: change the font. At the end of the references sections, there is a reference without being numbered.